# Contextual Combinatorial Multi-output GP Bandits with Group Constraints

**Sepehr Elahi**[†*]                                                              *sepehr.elahi@epfl.ch*
*School of Computer and Communication Sciences*
*EPFL*

**Baran Atalar**[†*]                                                              *batalar@andrew.cmu.edu*
*Department of Electrical and Computer Engineering*
*Carnegie Mellon University*

**Sevda Öğüt**[†*]                                                                *sevda.ogut@epfl.ch*
*School of Computer and Communication Sciences*
*EPFL*

**Cem Tekin**                                                                     *cemtekin@ee.bilkent.edu.tr*
*Department of Electrical and Electronics Engineering*
*Bilkent University*

**Reviewed on OpenReview:** *https://openreview.net/forum?id=OqbGu3hdQb*

## Abstract

In federated multi-armed bandit problems, maximizing global reward while satisfying minimum privacy requirements to protect clients is the main goal. To formulate such problems, we consider a combinatorial contextual bandit setting with groups and changing action sets, where similar base arms arrive in groups and a set of base arms, called a super arm, must be chosen in each round to maximize super arm reward while satisfying the constraints of the rewards of groups from which base arms were chosen. To allow for greater flexibility, we let each base arm have two outcomes, modeled as the output of a two-output Gaussian process (GP), where one outcome is used to compute super arm reward and the other for group reward. We then propose a novel double-UCB GP-bandit algorithm, called Thresholded Combinatorial Gaussian Process Upper Confidence Bounds (TCGP-UCB), which balances between maximizing cumulative super arm reward and satisfying group reward constraints and can be tuned to prefer one over the other. We also define a new notion of regret that combines super arm regret with group reward constraint satisfaction and prove that TCGP-UCB incurs $\tilde{O}(\sqrt{\lambda^*(K)KT\overline{\gamma}_T})$ regret with high probability, where $\overline{\gamma}_T$ is the maximum information gain associated with the set of base arm contexts that appeared in the first $T$ rounds and $K$ is the maximum super arm cardinality over all rounds. We lastly show in experiments using synthetic and real-world data and based on a federated learning setup as well as a content-recommendation one that our algorithm performs better then the current non-GP state-of-the-art combinatorial bandit algorithm, while satisfying group constraints.

## 1 Introduction

The multi-armed bandit problem is a prominent example of reinforcement learning that studies the sequential interaction of a learner with its environment under partial feedback (Robbins, 1952). Out of the many

---

[*]The majority of the work was done when Sepehr Elahi, Baran Atalar and Sevda Öğüt were with the Department of Electrical and Electronics Engineering of Bilkent University.

[†]Equal contribution.

variants of the multi-armed bandit problem, the combinatorial and contextual bandits have been thoroughly investigated due to their rich set of real-life applications. In combinatorial bandits, the learner selects a super arm, which is a subset of the available base arms in each round (Chen et al., 2013; Cesa-Bianchi & Lugosi, 2012). In the semi-bandit feedback model, at the end of the round, the learner observes the outcomes of the base arms in the selected super arm together with the super arm reward. In contextual bandits, at the beginning of every round, the learner observes side-information about the outcomes of the base arms available in that round before deciding which arm to select (Slivkins, 2011; Lu et al., 2010). In the changing action set variant, there is varying base arm availability in each round. Combinatorial bandits and contextual bandits are deployed in applications ranging from influence maximization in a social network to news article recommendation (Li et al., 2010; Chen et al., 2016b).

In this paper, we consider contextual combinatorial multi-armed bandits with changing action sets (C3-MAB), which models the repeated interaction between an agent and its dynamically changing environment. In each round $t$, the agent observes the available base arms and their contexts, selects a subset of the available base arms, which is called a super arm, collects a reward, and observes noisy outcomes of the selected base arms. We consider the scenario in which the base arm availability in each round changes in an arbitrary fashion, known as the changing action sets setting. Therefore, we analyze the regret under any given sequence of base arm availabilities. On the other hand, given a particular base arm and its context, the outcome of the base arm is assumed to come from a fixed distribution parameterized by the arm's context. The goal is to maximize the cumulative reward in a given number of rounds without knowing future context arrivals and the function that maps the contexts to base arm outcomes. Achieving this goal requires careful tuning of exploration and exploitation by adapting decisions in real-time based on the problem structure and past history. What is new in our setup is the concept of groups, which are unions of base arms that share a common feature. Our goal is to maximize the cumulative super arm reward while ensuring that the groups comprising the super arm satisfy their constraints.

It is essential to place assumptions on the function mapping contexts to outcomes (i.e., the function that needs to be learned, say $f$). While other C3-MAB works such as (Nika et al., 2020; Chen et al., 2016a; 2018) place explicit smoothness restrictions on $f$, we make use of Gaussian processes (GP), which induce a smoothness assumption. Moreover, the posterior mean and variance resulting from GP gives us high probability confidence bounds on the expected outcome function which leads to a desirable balance between exploration and exploitation (Srinivas et al., 2012).

Our work, and specifically introduction of groups, is motivated by the need to balance between privacy and reward in a federated learning setup. Federated learning (FL) is a distributed machine learning method that allows for training a model across a decentralized ensemble of clients. In FL, clients train a model on their local data and then send their local model to a centralized server. This helps protect the privacy of clients as they do not share their data directly with the server (McMahan et al., 2017). Our work introduces a setup that can handle a privacy aware FL, where clients have privacy requirements on the maximum amount of data leakage through local model or weight transfer to the server. This is important, as some portions of a client's training data can be retrieved just from their shared gradients with the server (Melis et al., 2019; Chen & Campbell, 2021).

In our setup, base arms represent client-request pairs, groups are clients, and super arms are subsets of client-request pairs. For each such pair, the context is related to the amount of information that the client is to share with the server, specifically, it is the used data set percentage which determines the privacy level of the client. Our aim is to satisfy clients' privacy requirements while maximizing the information obtained from the client's trained model. There is a trade-off between these two goals as using a higher percentage of data in training leads to less privacy for clients due to information leakage but a greater amount of information is retrieved from the data by the server. This is also mentioned in (Truong et al., 2021) as a trade-off between efficiency and privacy guarantee and in (Yang et al., 2019) as a trade-off between learning performance and privacy of clients.

This trade-off has motivated us to model the expected outcome function as a sample from a two-output GP, in order to make use of the inherent correlation between privacy leakage and the information retrieved. The

details of how a two-output GP helps us as well as a detailed explanation of the application of our problem formulation to a privacy aware federated learning is given in Section 2.8.

## 1.1 Our contributions

Our contributions are: **(i)** We propose a new C3-MAB problem with groups and group constraints that is applicable to real-life problems, including a privacy-aware federated learning setup where the goal is to maximize overall reward while respecting users' varying privacy requirements. **(ii)** We propose a new notion of regret called group regret, and define total regret as a weighted combination of group and super arm regret in terms of $\zeta \in [0, 1]$. **(iii)** We use a two-output GP which allows for modeling setups where overall reward and group reward are functions of two different but correlated base arm outcomes. **(iv)** We use a double UCB approach in our algorithm by having different exploration bonuses for groups and super arms which is dependent on a tunable parameter. **(v)** We derive an information theoretic regret bound for our algorithm given by $\tilde{O}(\sqrt{\lambda^*(K)KT\overline{\gamma}_T})$. By assuming a fixed cardinality for every super arm, we express our regret bound in terms of the classical maximum information gain $\gamma_T$ which is given as $\tilde{O}(\sqrt{\lambda^*(K)KT\gamma_T})$. We also provide kernel-dependent upper bounds of our regret for standard GP kernels.

## 1.2 Related Work

The combinatorial multi-armed bandit problem has been thoroughly investigated by using upper confidence bounds (Gai et al., 2012; Chen et al., 2013; 2016b) and Thompson sampling algorithms (Wang & Chen, 2018). While we also use a UCB type algorithm, we have groups and group constraints and we operate in a contextual combinatorial setting instead of a solely combinatorial one. In terms of the regret bounds, (Chen et al., 2013; 2016b; Wang & Chen, 2018) incur $O(\log T)$ gap-dependent regret whereas we incur $\tilde{O}(\sqrt{T})$ gap-independent regret. Another variant of the classical bandit problem which emerged in recent years is the contextual combinatorial multi-armed bandit problem (CC-MAB) (Li et al., 2016; Qin et al., 2014). In the work of (Li et al., 2016), the learner can observe the reward of the super arm and the rewards of a subset of the base arms in the super arm selected due to cascading feedback according to some stopping criteria. In the work of (Qin et al., 2014), the contextual combinatorial MAB problem is applied to online recommendation.

The contextual combinatorial multi-armed bandits with changing action sets problem has been explored in (Chen et al., 2018; Nika et al., 2020). In the work of (Chen et al., 2018), the super arm reward is submodular and since the context space is infinite, they form a partition of the context space with hypercubes depending on context information thus addressing the varying availability of arms by exploiting the similarities between contexts in the same hypercube. The work done by (Nika et al., 2020) makes use of adaptive discretization instead of fixed discretization which addresses the limited similarity information of the arms that can be gathered by fixed discretization.

GP bandits have been utilized in the combinatorial MAB and contextual MAB setup as well as for adaptive discretization (Nuara et al., 2018; Krause & Ong, 2011; Shekhar et al., 2018). In the combinatorial MAB setting of (Nuara et al., 2018), it has been shown that discretization works in small action spaces and GPs are used to estimate the model. While adaptive discretization has been shown to be working well in large context spaces and setups with changing action sets, in the case where the number of base arms is finite, assuming the expected base arm outcomes are a sample from a GP could be a better approach as it eliminates the need for the explicit assumption that the expected base arm outcomes are Lipschitz continuous. In this paper we also use the smoothness induced by a GP instead of performing explicit discretization. A comparison of our problem setup with other similar works can be found in Table 1.

The thresholding multi-armed bandit problem has been studied in (Locatelli et al., 2016; Mukherjee et al., 2017). These works consider thresholding as having the mean of a base arm be above a certain value. In (Reverdy et al., 2017), there is the notion of satisficing instead of thresholding, which is a combination of satisfaction, which is the learner's desire to have a reward above a threshold and sufficiency, meaning to have satisfaction for base arms at a certain level of confidence. To our knowledge, there is no previous research which deals with group thresholding or a thresholding setup within the C3-MAB problem. A similar line of research includes that of safety in the multi-armed bandit problem where the action set is constrained such that a safety constraint is satisfied (Amani et al., 2019; Wang et al., 2021). However, their definition of safety

Table 1: Comparison of our problem setting with previous works (CC stands for contextual-changing action sets).

| Work | Context space | CC | Smoothness | Groups | Group thresholds |
|------|--------------|----|-----------|--------|-----------------|
| (Chen et al., 2013) | Finite | No | Explicit | No | No |
| (Chen et al., 2018) | Infinite | Yes | Explicit | No | No |
| (Krause & Ong, 2011) | Compact | No | GP-induced | No | No |
| (Nika et al., 2020) | Compact | Yes | Explicit | No | No |
| This work | Compact | Yes | GP-induced | Yes | Yes |

is different than the notion of satisfying that we use in the paper. In these papers, the safety constraint is imposed such that it allows for no violation of the constraint whereas we can violate our threshold in some steps, which leads us to incur regret. The tradeoff however is that we are able to explore faster compared to these works though it is not fair to make a comparison as our setup and results are not exactly comparable due to different notions of thresholding and also because we are not doing best arm identification. Comparing with the regret bounds of (Amani et al., 2019) they achieve $\tilde{O}(\sqrt{T})$ regret (same as ours) when the safety gap is positive and $\tilde{O}(T^{2/3})$ regret when the safety gap is zero.

Federated learning was proposed by (McMahan et al., 2017) as a technique to address privacy and security concerns regarding centralized training performed in a main server by collecting private data from the clients. To solve this issue, they propose the FedAvg algorithm that performs local stochastic gradient descent for each client and averages the models from the clients on the main server to update the global model. The work in (Wei et al., 2020) discusses information leakage in federated learning and presents the algorithm NbAFL that works based on the concept of differential privacy by adding artificial noises to the parameters of the clients before aggregating the model. Note that although our setup is motivated by privacy aware FL, our work is not positioned in the FL space and thus our algorithm competes with C3-MAB algorithms and not the likes of NbAFL.

## 2 Problem Formulation

### 2.1 Base Arms and Base Arm Outcomes

The sequential decision-making problem proceeds over $T$ rounds indexed by $t \in [T] := \{1, \ldots, T\}$. In each round $t$, $M_t$ base arms indexed by the set $\mathcal{M}_t = [M_t]$ arrive. Cardinality of the set of available base arms in any round is bounded above by $M < \infty$. Each base arm $m \in \mathcal{M}_t$ comes with a context $x_{t,m}$ that resides in the context set $\mathcal{X}$. The set of available contexts in round $t$ is denoted by $\mathcal{X}_t = \{x_{t,m}\}_{m \in \mathcal{M}_t}$.

When selected, a base arm with context $x$ yields a two-dimensional random outcome $\boldsymbol{r}(x) \in \mathbb{R}^2$. The two-dimentionality of the outcome will be motivated by groups in Section 2.2. We will refer to the first and second component of $\boldsymbol{r}(x)$ by $r_1(x)$ and $r_2(x)$, respectively. From hereon, we refer to the individual components of a vector using subscripts. Then, the expected outcome function that is unknown to the learner is represented by $\boldsymbol{f} \colon \mathcal{X} \to \mathbb{R}^2$, whose form will be described in detail in Section 2.5. We then define the random outcome as $\boldsymbol{r}(x) = \boldsymbol{f}(x) + \boldsymbol{\eta}$ where $\boldsymbol{\eta} \sim \mathcal{N}(\boldsymbol{0}, \sigma^2 \boldsymbol{I})$ represents the two-dimensional observation noise that is independent across base arms and rounds, where $\sigma$ is known. In our motivating example of federated learning, base arms represent client-request pairs while the first base arm outcome is the amount of information that the server is able to retrieve from the client while the second one is the information leakage probability associated with each client-request pair. The context associated with each base arm corresponds to its data set usage percentage.

### 2.2 Super Arm and Group Rewards

For any vector-valued function $\boldsymbol{h} \colon \mathcal{X} \to \mathbb{R}^d$, $d \geq 2$, given a $k$-tuple of contexts $\boldsymbol{x} = [x_1, \ldots, x_k]$, we let $\boldsymbol{h}(\boldsymbol{x}) = [\boldsymbol{h}(x_1), \ldots, \boldsymbol{h}(x_k)]$. Moreover, we let $h_i(\boldsymbol{x}) = [h_i(x_1), \ldots, h_i(x_k)]$, where the subscript $1 \leq i \leq d$

indicates the $i$th element. We denote by $\mathcal{S}_t \in 2^{\mathcal{M}_t}$ the set of feasible super arms in round $t$, which is given to the learner, and by $\mathcal{S} = \cup_{t \geq 1} \mathcal{S}_t$ the overall feasible set of super arms. We assume that the maximum number of base arms in a super arm does not exceed a fixed $K \in \mathbb{N}$ that is known to the learner. That is, for any $S \in \mathcal{S}$, we have, $|S| \leq K$. We denote the super arm chosen in round $t$ by $S_t$.

The reward of a super arm $S \in \mathcal{S}$ depends on the outcomes of base arms in $S$. We represent the contexts of the base arms in super arm $S$ by the context vector $\boldsymbol{x}_{t,S}$ and the reward of a super arm $S$ by random variable $U(S, r_1(\boldsymbol{x}_{t,S}))$, where $U$ is a deterministic function. Moreover, we have that the expected super arm reward is a function of only the set of base arms in $S$ and their expected outcome vector, given by $u(S, f_1(\boldsymbol{x}_{t,S})) = \mathbb{E}[U(S, r_1(\boldsymbol{x}_{t,S}))|\boldsymbol{f}]$.[1] We assume that for all $S \in \mathcal{S}$, the expected super arm reward function $u$ is monotonically non-decreasing with respect to the expected outcome vector. We also assume that $u$ varies smoothly as a function of expected base arm outcomes. Both assumptions are standard in the C3-MAB literature (Nika et al., 2020) and are stated formally below.

**Assumption 1** (Monotonicity). *For all $S \in \mathcal{S}$ and for any $\boldsymbol{h} = [h_1, \ldots, h_{|S|}]^T \in \mathbb{R}^{|S|}$ and $\boldsymbol{g} = [g_1, \ldots, g_{|S|}]^T \in \mathbb{R}^{|S|}$, if $h_m \leq g_m$, $\forall m \leq |S|$, then $u(S, \boldsymbol{h}) \leq u(S, \boldsymbol{g})$.*

**Assumption 2** (Lipschitz continuity). *For all $S \in \mathcal{S}$, there exists $B' > 0$ such that for any $\boldsymbol{h} = [h_1, \ldots, h_{|S|}]^T \in \mathbb{R}^{|S|}$ and $\boldsymbol{g} = [g_1, \ldots, g_{|S|}]^T \in \mathbb{R}^{|S|}$, we have $|u(S, \boldsymbol{h}) - u(S, \boldsymbol{g})| \leq B' \sum_{i=1}^{|S|} |h_i - g_i|$.*

Groups are defined as the sets of base arms (i.e., client-request pairs) from the same client and they have round-varying cardinalities due to changing action sets. We denote by $\mathcal{G}_t$ the set of feasible groups in round $t$ and the reward of a group $G \in \mathcal{G}_t$ depends on the second outcome of the base arms in $G \cap S_t$. Let $\boldsymbol{x}_{t,G}$ represent the vector of contexts of base arms in $G$.[2] We only have one restriction on the set of arriving groups, and that is disjointness, given below.

**Assumption 3** (Group disjointness). *For any round $t$, we have that all groups in $\mathcal{G}_t$ are pairwise disjoint.*

We represent the reward of a group $G$ by the random variable $V_G(G \cap S_t, r_2(\boldsymbol{x}_{t,G \cap S_t}))$. Notice that the reward of a group depends on the observed base arms since the observed base arm outcomes in a group are ones that belong to the chosen super arm. We stress that $V_G$ here is a deterministic function of its argument and the randomness comes from $r_2$. As typical in the CMAB literature, given the base arm outcome function $\boldsymbol{f}$, we assume that expected group reward is a function of only the set of base arms in $G \cap S_t$ and their expected outcome vector. Therefore, we define $v_G(G \cap S_t, f_2(\boldsymbol{x}_{t,G \cap S_t})) = \mathbb{E}[V_G(G \cap S_t, r_2(\boldsymbol{x}_{t,G \cap S_t}))|\boldsymbol{f}]$. Note that $V_G$ and $v_G$ are vector functions and take as input a $|G \cap S_t|$ dimensional vector.[3] Lastly, similar to the super arm reward function, we impose monotonicity and Lipschitz continuity assumptions on the group reward function.

**Assumption 4** (Monotonicity). *For all $A \subseteq G$, for all $G \in \mathcal{G}_t$, and for all $t \in [T]$ and for any $\boldsymbol{h} = [h_1, \ldots, h_{|A|}]^T \in \mathbb{R}^{|A|}$ and $\boldsymbol{g} = [g_1, \ldots, g_{|A|}]^T \in \mathbb{R}^{|A|}$, if $h_m \leq g_m$, $\forall m \leq |A|$, then $v_G(\boldsymbol{h}) \leq v_G(\boldsymbol{g})$.*

**Assumption 5** (Lipschitz continuity). *For all $A \subseteq G$, for all $G \in \mathcal{G}_t$ and for all $t \in [T]$, there exists $B_A > 0$ such that for any $\boldsymbol{h} = [h_1, \ldots, h_{|A|}]^T \in \mathbb{R}^{|A|}$ and $\boldsymbol{g} = [g_1, \ldots, g_{|A|}]^T \in \mathbb{R}^{|A|}$, we have $|v_G(\boldsymbol{h}) - v_G(\boldsymbol{g})| \leq B_A \sum_{i=1}^{|A|} |h_i - g_i|$.*

We define $B := \max_{\forall A \subseteq G, \forall G \in \mathcal{G}_t} \max_{\forall t \in [T]} B_A$, which will be later used in proofs. In federated learning, group reward is related to the information leakage probability of each base arm in a group. Therefore, it represents the privacy level of a client.

In line with prior work (Nika et al., 2021), we assume that the learner knows $u$ and $v_G$ for all $G$ perfectly, but does not know $\boldsymbol{f}$ beforehand. We propose an extended semi-bandit feedback model, where when super arm $S_t$ is selected in round $t$, the learner observes $U(r_1(\boldsymbol{x}_{S_t}))$, $V_G(r_2(\boldsymbol{x}_{G \cap S_t}))$ for $G \in \mathcal{G}_t$ and $\boldsymbol{r}(x_{t,m})$ for $m \in S_t$ at the end of the round.

---

[1]We will suppress $S$ from $U(S, \ldots)$ and $u(S, \ldots)$ when $S$ is clear from context.

[2]We will drop $t$ from $\boldsymbol{x}_{t,G}$ when clear from context.

[3]We will suppress $G \cap S_t$ from $V_G(G \cap S_t, \ldots)$ and $v_G(G \cap S_t, \ldots)$ when clear from context.

### 2.3 Regret

Our learning objective is to choose super arms that yield maximum rewards while ensuring that the groups containing the base arms of the chosen super arms satisfy a certain level of quality, which is characterized by the group threshold. In accordance with our goal, we define regret notions for both super arms and groups. In our regret analysis, as typical in contextual bandits (Nika et al., 2020), we assume that the sequence $\{\mathcal{X}_t, \mathcal{S}_t, \mathcal{G}_t\}_{t=1}^{T}$ is fixed, hence context arrivals are not affected by past actions. We assume that each group $G \in \mathcal{G}_t$ has a threshold $\gamma_{t,G}$, which represents the minimum reward that the group requires when one or more of its forming base arms are in the chosen super arm. For instance, in federated learning, $\gamma_{t,G}$ represents the minimum desired privacy level of each client, specifically the probability that the client's data does not leak. This is important as we want to maintain privacy for clients during training. We say that group $G \in \mathcal{G}_t$ satisfies its threshold when $v_G(f_2(\boldsymbol{x}_{G \cap S_t})) \geq \gamma_{t,G}$. We then define the group regret as

$$R_g(T) = \sum_{t=1}^{T} \sum_{G \in \mathcal{G}_t} [\gamma_{t,G} - v_G(f_2(\boldsymbol{x}_{G \cap S_t}))]_+,$$

where $[\cdot]_+ := \max\{\cdot, 0\}$.

Let $\mathcal{S}'_t \subseteq \mathcal{S}_t$ represent the set of super arms whose forming base arms' membering groups satisfy their thresholds, mathematically given by $\mathcal{S}'_t := \{S \in \mathcal{S}_t : \forall(G \in \mathcal{G}_t), v_G(f_2(\boldsymbol{x}_{t,G \cap S})) \geq \gamma_{t,G}\}$. The super arm regret is defined as the standard $\alpha$-approximation regret in CMAB:

$$R_s(T) = \alpha \sum_{t=1}^{T} \mathrm{opt}(f_t) - \sum_{t=1}^{T} u(f_1(\boldsymbol{x}_{t,S_t})),$$

where $\mathrm{opt}(f_t) = \max_{S \in \mathcal{S}'_t} u(f_1(\boldsymbol{x}_{t,S}))$. For this definition to make sense, we impose the following assumption.

**Assumption 6** (At least one good super arm). *Given a round t, we have that $\mathcal{S}'_t \neq \emptyset$.*

The assumption above is a reasonable assumption for a real-world scenario, as we are simply requiring that there exists at least one super arm whose forming base arms satisfy their respective groups. Further note that any optimal super arm $S^*_t \in \arg\max_{S \in \mathcal{S}'_t} u(f_1(\boldsymbol{x}_{t,S}))$ is restricted to satisfy all its corresponding groups and this will be further explained in Section 2.4.2. Otherwise, the policy which always selects the optimal super arms will incur linear group regret.

The total regret is defined as a weighted combination of group and super arm regrets. We have been inspired by the scalarization approach (Boyd & Vandenberghe, 2004) which has been used in multi-objective reinforcement learning (Van Moffaert et al., 2013) and in multi-objective multi armed bandits (Yahyaa et al., 2014). We use a similar regret definition to that in (Yahyaa et al., 2014) where they use scalarized regret for multi-objectives. This is also applicable to our case as we want to balance between prioritizing super arm reward and group reward. Given the trade-off parameter $\zeta \in [0, 1]$, we define it as

$$R(T) = \zeta R_g(T) + (1 - \zeta)R_s(T) . \tag{1}$$

Setting $\zeta = 1$ reduces the problem to satisfying problem in a combinatorial setup, while setting $\zeta = 0$ reduces the problem to regret minimization in standard CMAB (Chen et al., 2013).

### 2.4 Computation Oracle

In the traditional C3-MAB setting where group constraints are not considered, there exists one optimization problem, namely identifying a super arm to play in each round (Chen et al., 2016a; Nika et al., 2020). However, in our new setup where group constraints are considered, an additional optimization problem needs to be solved to identify super arms whose comprising groups i.e., the groups that the base arms of the selected super arm belong to satisfy their constraints. To do so, first we identify *good* subgroups and then filter out the super arms whose intersections with groups is not a good subgroup.

### 2.4.1 Identifying Good Subgroups

We define a good subgroup of $G$ to be any subset of $G$ that satisfies $G$'s constraint. For instance, given $G = \{1, 2, 3, 4\}$, $\gamma_G = 6$, and $v_G$ being the sum operator, then the following subsets are all good subgroups as they all satisfy the constraint: $\{2, 4\}, \{3, 4\}, \{1, 2, 4\}, \{2, 3, 4\}, \{1, 3, 4\}, \{1, 2, 3, 4\}$. Then, given that $\hat{f}_{t,2}$ is the estimate of the base arm outcome used for computing group reward, $f_2$, in round $t$, the good subgroup identification problem in round $t$ can be formally expressed as identifying the set $\mathcal{G}_{t,\text{good}}$ defined as $\mathcal{G}_{t,\text{good}} := \{G' \subseteq G \mid G \in \mathcal{G}_t \text{ and } v_G(\hat{f}_{t,2}(\boldsymbol{x}_{G'})) \geq \gamma_{t,G'}\}$. Thus, in round $t$, we have $\mathcal{G}_{t,\text{good}} = \text{Oracle}_{\text{grp}}(\hat{f}_{t,2})$.

### 2.4.2 Identifying the Optimal Super Arm

Once $\mathcal{G}_{t,\text{good}}$ is identified, the next task is to identify the super arm that yields the highest reward. Using the set of good subgroups returned by $\text{Oracle}_{\text{grp}}$, the optimization problem of finding the optimal super arm in round $t$, $S_t$, given the estimate of the base arm outcome used for computing super arm reward, $\hat{f}_{t,1}$, can be written as $S_t = \arg\max_S\{u(\hat{f}_{t,1}(\boldsymbol{x}_{t,S})) \mid S \in \mathcal{S}_t \text{ and } \forall G \in \mathcal{G}_t, \ S \cap G \in \mathcal{G}_{t,\text{good}}\}$. In other words, $S_t$ is the super arm that yields the highest reward calculated using $\hat{f}_{t,1}$ and whose intersections with any group is a good subgroup (i.e., satisfies that group's constraints). Note that if $\mathcal{G}_{t,\text{good}}$ is computed to be $\emptyset$, then $S_t$ is simply the super arm that yields the highest reward. Then, we make use of an $\alpha$-approximate oracle, called $\text{Oracle}_{\text{spr}}$, where $u(\hat{f}_{t,1}(\boldsymbol{x}_{t,\text{Oracle}_{\text{spr}}(\hat{f}_{t,1})})) \geq \alpha \times \text{opt}(\hat{f}_{t,1})$. Our algorithm, which will be described in detail in Section 3, will make use of both oracles. Lastly, it should be noted that both oracles are deterministic given their inputs.

## 2.5 Structure of Base Arm Outcomes

To ensure that the learner performs well, some regularity conditions are necessary on $\boldsymbol{f}$. For our paper, we model $\boldsymbol{f}$ as a sample from a two-output GP, defined below.

**Definition 1.** *A two-output Gaussian Process with index set $\mathcal{X}$ is a collection of 2-dimensional random variables $(\boldsymbol{f}(x))_{x \in \mathcal{X}}$ which satisfy the condition that $(\boldsymbol{f}(x_1), \dots, \boldsymbol{f}(x_n))$ has a multivariate normal distribution for all $(x_1, \dots, x_n)$ and $n \in \mathbb{N}$. The probability law of the GP is governed by its vector-valued mean function given by $x \mapsto \boldsymbol{\mu}(x) = \mathbb{E}[\boldsymbol{f}(x)] \in \mathbb{R}^2$ and its matrix-valued covariance function given by $(x_1, x_2) \mapsto \boldsymbol{k}(x_1, x_2) = \mathbb{E}[(\boldsymbol{f}(x_1) - \boldsymbol{\mu}(x_1))(\boldsymbol{f}(x_2) - \boldsymbol{\mu}(x_2))^T] \in \mathbb{R}^{2 \times 2}$.*

We assume that we have bounded variance, that is, $k_{jj}(x, x) \leq 1$ for every $x \in \mathcal{X}$ and $j \in \{1, 2\}$. This is a standard assumption generally used in GP bandits (Srinivas et al., 2012).

## 2.6 Posterior Distribution of Base Arm Outcomes

Our learning algorithm will make use of the posterior distribution of the two-output GP-sampled function $\boldsymbol{f}$. Given a fixed $N \in \mathbb{N}$ we consider a finite sequence $\tilde{\boldsymbol{x}}_{[N]} = [\tilde{x}_1, \dots, \tilde{x}_N]^T$ of contexts with corresponding outcome vector ($2N$-dimensional) $\boldsymbol{r}_{[N]} := [\boldsymbol{r}(\tilde{x}_1)^T, \dots, \boldsymbol{r}(\tilde{x}_N)^T]^T$ and the corresponding expected outcome vector ($2N$-dimensional) $\boldsymbol{f}_{[N]} = [\boldsymbol{f}(\tilde{x}_1)^T, \dots, \boldsymbol{f}(\tilde{x}_N)^T]^T$. For every $n \leq N$, we have $\boldsymbol{r}(\tilde{x}_n) = \boldsymbol{f}(\tilde{x}_n) + \boldsymbol{\eta}_n$ where $\boldsymbol{\eta}_n$ is the noise corresponding to that outcome. The posterior distribution of $\boldsymbol{f}$ given $\boldsymbol{r}_{[N]}$ is that of a two-output GP characterized by its mean $\boldsymbol{\mu}_N$ and its covariance $\boldsymbol{k}_N$ which are given as follows:

$$\boldsymbol{\mu}_N(\tilde{x}) = (\boldsymbol{k}_{[N]}(\tilde{x}))(\boldsymbol{K}_{[N]} + \sigma^2 \boldsymbol{I}_{2N})^{-1} \boldsymbol{r}_{[N]}^T,$$
$$\boldsymbol{k}_N(\tilde{x}, \tilde{x}') = \boldsymbol{k}(\tilde{x}, \tilde{x}') - (\boldsymbol{k}_{[N]}(\tilde{x}))(\boldsymbol{K}_{[N]} + \sigma^2 \boldsymbol{I}_{2N})^{-1} \boldsymbol{k}_{[N]}(\tilde{x}')^T.$$

Here $\boldsymbol{k}_{[N]}(\tilde{x}) = [\boldsymbol{k}(\tilde{x}, \tilde{x}_1), \dots, \boldsymbol{k}(\tilde{x}, \tilde{x}_N)] \in \mathbb{R}^{2 \times 2N}$ and

$$\boldsymbol{K}_{[N]} = \begin{bmatrix} \boldsymbol{k}(\tilde{x}_1, \tilde{x}_1), & \cdots, & \boldsymbol{k}(\tilde{x}_1, \tilde{x}_N) \\ \vdots & \ddots & \vdots \\ \boldsymbol{k}(\tilde{x}_N, \tilde{x}_1) & \cdots, & \boldsymbol{k}(\tilde{x}_N, \tilde{x}_N) \end{bmatrix}$$

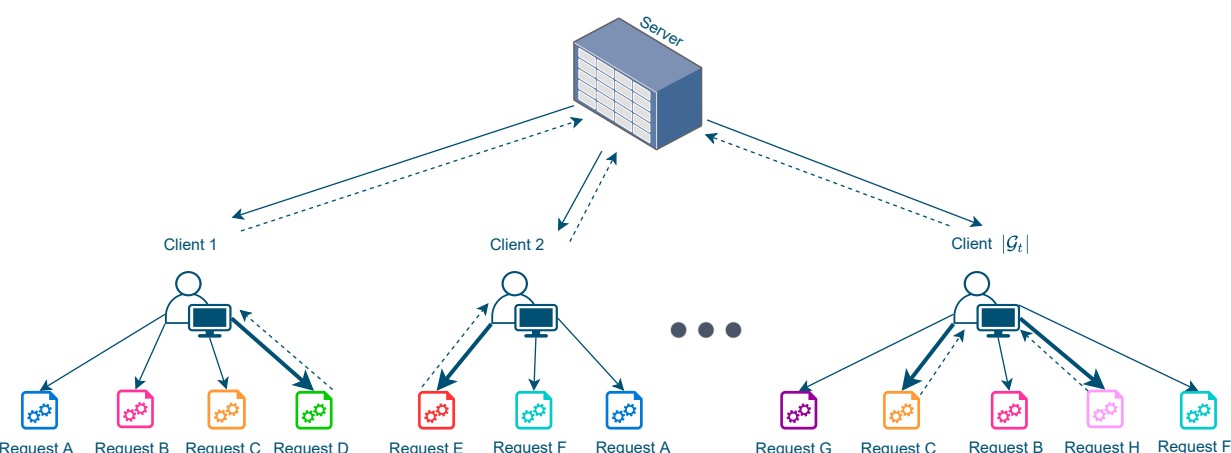

Figure 1: Illustration of a privacy-aware federated learning setup. A central server can offer different clients, who have varying privacy requirements, tasks that require the client to train a model using a given portion of their local data set.

Overall, the posterior of $\boldsymbol{f}(x)$ is given by $\mathcal{N}(\boldsymbol{\mu}_N(x), \boldsymbol{k}_N(x, x))$ and for each $j \in \{1, 2\}$, the posterior distribution of $f_j(x)$ is $\mathcal{N}(\mu_{j\,N}(x), (\sigma_{j\,N}(x))^2)$, where $(\sigma_{j\,N}(x))^2 = k_{jj\,N}(x, x)$. Moreover, the posterior distribution of the outcome $\boldsymbol{r}(x)$ is given by $\mathcal{N}(\boldsymbol{\mu}_N(x), \boldsymbol{k}_N(x, x) + \sigma^2 \boldsymbol{I}_2)$.

## 2.7 The Information Gain

Regret bounds for GP bandits depend on how well $\boldsymbol{f}$ can be learned from sequential interaction. The learning difficulty is quantized by the information gain, which accounts for the reduction in entropy of a random vector after a sequence of (correlated) observations. For a length $N$ sequence of contexts $\tilde{\boldsymbol{x}}_{[N]}$, evaluations of $\boldsymbol{f}$ at contexts in $\tilde{\boldsymbol{x}}_{[N]}$, given by $\boldsymbol{f}_{[N]}$, and the corresponding sequence of outcomes $\boldsymbol{r}_{[N]}$, the information gain is defined as $I(\boldsymbol{r}_{[N]}; \boldsymbol{f}_{[N]}) := H(\boldsymbol{r}_{[N]}) - H(\boldsymbol{r}_{[N]}|\boldsymbol{f}_{[N]})$ where $H(\cdot)$ and $H(\cdot|\cdot)$ represent the entropy and the conditional entropy operators, respectively. Essentially, the information gain quantifies the reduction in the entropy of $\boldsymbol{f}_{[N]}$ given $\boldsymbol{r}_{[N]}$. The maximum information gain over the context set $\mathcal{X}$ is denoted by $\gamma_{j\,N} := \sup_{\tilde{\boldsymbol{x}}_{[N]}:\tilde{x}_i \in \mathcal{X}, i \in [N]} I_j(\boldsymbol{r}_{[N]}; \boldsymbol{f}_{[N]})$. Similarly as before, $j \in \{1, 2\}$.

It is important to understand that $\gamma_{j\,N}$ is the maximum information gain for an $N$-tuple of contexts from $\mathcal{X}$. For large context spaces, $\gamma_{j\,N}$ can be very large. As we have varying base arm availability from round to round due to changing action sets, we only need to know the maximum information gain that is associated with a fixed sequence of base arm contexts arrivals. As we have a finite maximum information gain, this motivates us to relate the regret bounds to the informativeness of the available base arms. To adapt the definition of the maximum information gain to our setup with changing action sets and to ensure tight bounds, we define a new term $\overline{\gamma_{j\,T}}$. We let $\mathcal{Z}_t \subset 2^{\mathcal{X}_t}$ for $t \geq 1$ be the set of sets of context vectors which correspond to the base arms in the feasible set of super arms $\mathcal{S}_t$. We let $\tilde{\boldsymbol{z}}_t := \boldsymbol{x}_{t,S}$ be an element of $\mathcal{Z}_t$ for a super arm $S$ and we let $\tilde{\boldsymbol{z}}_{[T]} := [\tilde{\boldsymbol{z}}_1^T, \ldots, \tilde{\boldsymbol{z}}_T^T]$. Then the maximum information gain which is associated with the context arrivals $\mathcal{X}_1, \ldots, \mathcal{X}_T$ is

$$\overline{\gamma_{j\,T}} = \max_{\tilde{\boldsymbol{z}}_{[T]}:\tilde{\boldsymbol{z}}_t \in \mathcal{Z}_t, t \leq T} I(r_j(\tilde{\boldsymbol{z}}_{[T]}); f_j(\tilde{\boldsymbol{z}}_{[T]})).$$

The information gain for any $T$ element sequence of feasible super arms has a maximum given by $\overline{\gamma_{j\,T}}$. As it will be shown in Section 4, our regret bounds depend on $\overline{\gamma_{j\,T}}$.

### 2.8 Motivation

Our first motivation is a crowdsourcing setup, depicted in Figure 1. At each round $t$, $|\mathcal{G}_t|$ groups, hence clients, are available. Each client has one or more requests and these client-request pairs correspond to base arms, $m \in \mathcal{M}_t$. Super arms are comprised of combinations of these pairs. The chosen super arm $S_t$ includes the client-request pairs having bold arrows. These pairs also have dashed arrows as they train their data locally before sending their model back to the server. Base arm outcomes, namely $\boldsymbol{f}$, are the amount of information that the server retrieves and also the information leakage probability associated with each pair. The amount of retrieved information which is related to super arm reward $(u)$ is aimed to be maximized, and the information leakage probability which is inversely related to the group reward $(v_G)$ is aimed to satisfy a certain privacy threshold $(\gamma_{t,G})$. The context coming at each round $t$ for a base arm $m$, denoted by $x_{t,m}$, is the data set usage percentage for each client-request pair and hence affects base arm outcomes. The data set usage percentage affects information that the server retrieves since a higher percentage means the client is using more of its data to train the model and this percentage also affects the information leakage probability since training on larger portions of dataset leads to more information leaked as explored in (Melis et al., 2019). The server cares about the clients' privacy requirements as privacy is an important factor which influences the likelihood of a client enrolling in future crowdsourcing jobs. Essentially, a client is less likely to participate in future crowdsourcing jobs from a server if the server repeatedly requests the client to perform tasks which cause data leakage beyond the privacy leakage preference of the client. Also in this setup, it is not possible for clients to reject tasks which are assigned to them, however, if the server assigns tasks which disregard the privacy preference of the clients, the clients will likely become dissatisfied with the server and unlikely to use it for future jobs.[4] The clients leaving corresponds to changing arm sets in this model since when a client leaves its corresponding requests are also no longer available.

Another motivation for our setup is movie recommendation with content caching. In this formulation, base arms can be movie-user pairs and context can represent the alignment between the users' interests and the movies' contents, which is captured by the rating. Groups can be taken to be the users that are in the same location and super arms can again be the combination of various base arms. When a super arm is selected, movies of the base arms that are contained in this super arm are cached in their respective locations, hence, maximizing super arm reward is related to caching content that will be watched and liked by people. In this problem, we want to identify which movies to recommend and cache and in which locations we should cache these movies. Hence, we want to maximize the super arm reward which is a function of the ratings of the movies that we recommend to the users, in order to cache these movies since they are rated highly. But we also want to ensure that the movies that we cached at a certain location are well-liked on average by the users at that location since this is the main goal of caching. This corresponds to maximizing group reward. We give a more detailed explanation with mathematical notation for these two setups in Section 5.

## 3 Algorithm

We design an optimistic algorithm that uses GP UCBs in order to identify groups that satisfy thresholds and super arms that maximize rewards. We name our algorithm Thresholded Combinatorial GP-UCB (TCGP-UCB) with pseudo-code given in Algorithm 1. TCGP-UCB uses the *double UCB* principle, using different exploration bonuses for group and super arm selection. These bonuses are adjusted based on the trade-off parameter $\zeta$ given in the regret definition in equation 1.

For every available base arm, two indices which are upper confidence bounds on the two expected outcomes are defined by considering the parameter $\zeta \in [0, 1]$. Consider base arm $m$ with its associated context $x_{t,m}$. We define its reward index as

$$i_t(x_{t,m}) = \mu_{1\llbracket t-1 \rrbracket}(x_{t,m}) + \frac{1}{1-\zeta}(\sqrt{\beta_t})\sigma_{1\llbracket t-1 \rrbracket}(x_{t,m}),$$

---

[4]Note that our setup is general and we make no assumption on past base arm contexts affecting future ones.

where $\mu_{1[[t-1]]}(x_{t,m})$ and $\sigma_{1[[t-1]]}(x_{t,m})^2$ stand for the posterior mean and variance. Similarly, we define the satisfying index of base arm $m$ as

$$i'_t(x_{t,m}) = \mu_{2[[t-1]]}(x_{t,m}) + \frac{1}{\zeta}(\sqrt{\beta_t})\sigma_{2[[t-1]]}(x_{t,m}).$$

We give the indices $i'_t(x_{t,m})$ to Oracle$_{\text{grp}}$, which returns the set of good subgroups, $\mathcal{G}_{t,\text{good}}$. We form the set of feasible super arms whose groups are expected to satisfy their constraints, $\hat{\mathcal{S}}'_t$, by using the returned good subgroups and $\mathcal{S}_t$. If the oracle computes $\mathcal{G}_{t,\text{good}}$ to be the empty set, then we set $\hat{\mathcal{S}}'_t = \mathcal{S}_t$. Otherwise, we have that $\hat{\mathcal{S}}'_t = \{S \in \mathcal{S}_t : \forall G \in \mathcal{G}_t, \ S \cap G \in \mathcal{G}_{t,\text{good}}\}$. Then, we give the indices $i_t(x_{t,m})$ of the base arms that construct $\hat{\mathcal{S}}'_t$ to Oracle$_{\text{spr}}$, which returns an $\alpha$-optimal super arm.

---

**Algorithm 1** TCGP-UCB

1: **Input:** $\mathcal{X}, K, M, \zeta, \delta, T$; GP Prior: $\mathcal{GP}(\boldsymbol{\mu}_0, \boldsymbol{k})$
2: **for** $t = 1, \ldots, T$ **do**
3:     Observe base arms in $\mathcal{M}_t$, their contexts $\mathcal{X}_t$, their groups $\mathcal{G}_t$, and feasible super arms $\mathcal{S}_t$
4:     **for** $x_{t,m} : m \in \mathcal{M}_t$ **do**
5:         Calculate $\boldsymbol{\mu}_{[[t-1]]}(x_{t,m})$ and $\boldsymbol{\sigma}_{[[t-1]]}(x_{t,m})$ using GP posterior
6:         Compute indices $i_t(x_{t,m})$ and $i'_t(x_{t,m})$
7:     **end for**
8:     $\mathcal{G}_{t,\text{good}} \leftarrow \text{Oracle}_{\text{grp}}(i'_t(x_{t,m})_{m \in \mathcal{M}_t}, \mathcal{G}_t)$
9:     Form the set $\hat{\mathcal{S}}'_t$ using $\mathcal{G}_{t,\text{good}}$ and $\mathcal{S}_t$
10:     $S_t \leftarrow \text{Oracle}_{\text{spr}}(i_t(x_{t,m})_{m \in \hat{\mathcal{S}}'_t}, \hat{\mathcal{S}}'_t)$
11:     Observe base arm outcomes $\boldsymbol{r}$, collect group rewards $V_G$ and super arm reward $U$
12: **end for**

---

## 4  Theoretical Analysis

Next is our main result which asserts a high probability upper bound on the total regret of our algorithm in terms of $\overline{\gamma}_T$. The supplemental document contains further details and proofs of our results, most of which follow directly from those of (Nika et al., 2021). Throughout this section, we take $j \in \{1, 2\}$.

**Theorem 1.** *Super arm regret and group regret incurred by TCGP-UCB in $T$ rounds are upper bounded with probability at least $1 - \delta$ where $\delta \in (0, 1)$, $T \in \mathbb{N}$ and $\beta_t = 2\log(M\pi^2 t^2/3\delta)$ as follows:*

$$R_g(T) \leq \sqrt{C_1(K)\beta_T KT\overline{\gamma_{2T}}},$$

*where $C_1(K) = 2B^2(\frac{\zeta+1}{\zeta})^2(\lambda^*(K) + \sigma^2)$ and*

$$R_s(T) \leq \sqrt{C_2(K)\beta_T KT\overline{\gamma_{1T}}},$$

*where $C_2(K) = 2B'^2(\frac{2-\zeta}{1-\zeta})^2(\lambda^*(K) + \sigma^2)$. Hence, the total regret is bounded by:*

$$R(T) \leq \sqrt{C(K)\beta_T KT\overline{\gamma}_T},$$

*where $C(K) = 8(B + B')^2(\lambda^*(K) + \sigma^2)$, $\overline{\gamma}_T = \max\{\overline{\gamma_{1T}}, \overline{\gamma_{2T}}\}$, and $\lambda^*(K)$ is the maximum eigenvalue of $\Sigma_{[[t-1]]}(\boldsymbol{z}_t)$, for $t \leq T$ (i.e., maximum eigenvalue of all covariance matrices of selected actions).*

**Remark 1.** *Note the dependence of the bounds on the maximum eigenvalue of all covariance matrices throughout all rounds, denoted by $\lambda^*(K)$, which is a function of $K$. The worst cases upper bounds on $\lambda^*$ are known to be $O(K)$ (Zhan, 2005), in which case we obtain linear dependence on $K$ in the regret bounds. This dependence comes from the fact that base arm outcomes are not independent of each other. On the other hand, when the base arm outcomes are all independent, the covariance matrices in each round would be diagonal with $\lambda^* = \max_{t \leq T, k \leq K} \sigma^2_{j[[t-1]]}(x_{t,k}) \leq 1$, and thus, we incur regret bounds of the form $O(\sqrt{KT\overline{\gamma}_T})$.*

Next, we state lower and upper bounds on $\overline{\gamma}_T$ in terms of the classical information gain. This gives us a measure of how tight our bounds are with respect to previous bounds in the GP bandit literature.

**Theorem 2.** *Letting $\overline{T} = \sum_{t=1}^{T} |S_t|$ be the sum of cardinalities of selected actions up to time $T$, we have that*

$$\overline{\gamma}_T \leq \gamma_{\overline{T}} \ .$$

*Furthermore, if $|S_t| = K$ and $\mathcal{X}_t = \mathcal{X}$, for all $t \in [T]$, then we have*

$$\frac{1}{K}\gamma_{KT} \leq \overline{\gamma}_T \leq \gamma_{KT} \ .$$

Using this characterization of $\overline{\gamma}_T$, we can now provide a regret bound that depends on the classical notion of maximum information gain.

**Theorem 3.** *Fix $\delta \in (0,1)$ and let $T, K \in \mathbb{N}$. Under the conditions of Theorem 1 and assuming that $|S| = K$ for any $S \in \mathcal{S}$, let $\overline{T} = \sum_{t=1}^{T} |S_t|$. Then, the total regret incurred by TCGP-UCB in $T$ rounds is upper bounded with the following with probability at least $1 - \delta$:*

$$R(T) \leq \sqrt{C(K)\beta_T KT\gamma_{\overline{T}}},$$

*where*

$$\gamma_{\overline{T}} = \max_j \max_{A \subset \mathcal{X}: |A| = \overline{T}} I_j(r_j(\boldsymbol{z}_A); f_j(\boldsymbol{z}_A)) \tag{2}$$

*while $C$ and $\beta_T$ are the same as in Theorem 1.*

Finally, using kernel-dependent explicit bounds on $\gamma_{\overline{T}}$ given in (Srinivas et al., 2012; Vakili et al., 2020), we state a corollary of Theorem 3 that gives similar bounds on the total regret incurred by TCGP-UCB.

**Corollary 1.** *Let $\delta \in (0,1)$, $T, K \in \mathbb{N}$ and let $\mathcal{X} \subset \mathbb{R}^D$ be compact and convex. Under the conditions of Theorem 3 and for the following kernels, the total regret incurred by TCGP-UCB in $T$ rounds is upper bounded (up to polylog factors) with probability at least $1 - \delta$ as follows:*

1. *For the linear kernel we have: $R(T) \leq \tilde{O}\left(\sqrt{\lambda^*(K)DKT}\right)$.*

2. *For the RBF kernel we have: $R(T) \leq \tilde{O}\left(\lambda^*(K)\sqrt{DKT}\right)$.*

3. *For the Matérn kernel we have: $R(T) \leq \tilde{O}\left(\lambda^*(K)T^{(D+\nu)/(D+2\nu)}\right)$,*

*where $\nu > 1$ is the Matérn parameter.*

## 5 Experiments

We perform two sets of experiments. In the first one, we use a synthetic dataset and a privacy-aware federated learning setup to show that our algorithm outperforms the non-GP state-of-the-art while maintaining user privacy requirements. In the second experiment, we consider a content caching-aware movie recommendation setup. We also perform further simulations in Appendix B to investigate the effect of the trade-off parameter, $\zeta$.

### 5.1 Privacy-aware Federated Learning

#### 5.1.1 Setup

We consider a privacy-aware federated learning setup comprised of a server and clients. The server's goal is to train a model using clients' data. However, clients do not wish to directly share their data

with the server and instead train the model on a portion of their local dataset, with the portion decided by the server, which they then send to the server. Thus, in each round, the server must serve requests to the available clients and select the portion of their dataset to locally train on. Moreover, each client has different privacy requirements regarding the maximum amount of data that can be leaked.

In each round, we simulate the number of available clients using a Poisson distribution with mean 50. Then, the server can make a request to each client to ask them to use $x$ portion of their dataset for training the local model and then send the updated model to the server. We allow for $x \in \{0.01, 0.02, \dots, 1.00\}$. Thus, for each client there are 100 possible requests. We then represent each request-client pair as a base arm with a one-dimensional context in $[0, 1]$ that represents how much of the dataset the request asks for the client to use. We allow for the same client to receive multiple requests, as a client may have different local datasets on which he can train multiple local models. The maximum number of requests that each client can simultaneously receive in one round follow a Poisson distribution with mean 5.

After picking base arms (i.e., client-request pairs), the clients train the models on their local datasets and then return their model updates. We model the amount of useful information that the server gained from each request as a sigmoid-like function. More specifically, given context $x$, we have that the expected super arm outcome of $x$ is $f_1(x) = \frac{1}{1+\exp(5-10x)}$. This model is justified by observing that very little data leads to overfitting and hence the first flat region for small $x$, and once the amount of data surpasses some threshold, noticeable gain in the amount of useful information transferable from the local model can be seen, hence the linear portion of $f_1$. We define the random outcome to be $r_1(x) = f_1(x) + \eta_1$, where $\eta_1$ is zero-mean Gaussian noise with a standard deviation of 0.05. Lastly, as the amount of data keeps increasing, we have diminishing returns in the amount of useful information, captured by the flat region of $f_1$ for large $x$. Then, the super arm reward is the sum of all the information learned from all requests. Thus, $u(f_1(\boldsymbol{x}_S)) = \sum_{i=1}^{|S|} f_1(x_i)$.

We model each client's privacy requirement using a privacy leakage threshold in $[0, 1]$, where clients intend for the amount of information leaked from their local dataset to be below this threshold, where we sample the leakage threshold of each client uniformly at random from $[0, 1]$. When clients train on larger portions of their datasets, more of their training set and hence information is leaked (Melis et al., 2019). We model this leakage as $f_2(x) = 0.05 + 0.95 \exp(-5x)$. We justify this model by first noting that if very few training samples are used, then near-exact versions of the training samples can be extracted from the model updates or gradients (Melis et al., 2019). However, as the number of training samples increases, the leakage also decreases but never goes to 0. Additionally, we define the random group outcome to be $r_2(x) = f_2(x) + \eta_2$, where $\eta_2$ is zero-mean Gaussian noise with a standard deviation of 0.05. Finally, defining groups as set of client-request pairs from the same client, group reward–better called loss in this scenario–is the total leakage amount of all the requests of the client.[5]

### 5.1.2 Algorithms

We run the simulation using a slightly modified version of our algorithm, STCGP-UCB, and the non-GP C3-MAB state-of-the-art algorithm, ACC-UCB of (Nika et al., 2020).

**STCGP-UCB**

We run a slightly modified version of our algorithm that uses sparse approximation to GPs, called STCGP-UCB, where we use the sparse approximation to the GP posterior described in (Titsias, 2009). In this sparse approximation, instead of using all of the arm contexts up to round $t$ to compute the posterior, a small $s$ element subset of them is used, called the inducing points. By using a sparse approximation, the time complexity of the posterior updating procedure reduces from $O(K^3 + (2K)^3 + \dots + (KT)^3) = O(K^3T^4)$ to $O(s^2KT^2)$. In our simulation, we set $s = 10$. We also set $\delta = 0.05$, $\zeta = 0.5$, and use two squared exponential kernels with both lengthscale and variance set to 1.

---

[5]Even though group reward is defined as the non-leakage probability, considering it as the total leakage is also acceptable as our proofs still hold.

**ACC-UCB**

We set $v_1 = 1, v_2 = 1, \rho = 0.5$, and $N = 2$, as given in Definition 1 of (Nika et al., 2020). The initial (root) context cell, $X_{0,1}$, is a line centered at $(0.5)$.

### 5.1.3 Results

We run the simulation for 100 rounds with eight independent runs, averaging over the results of each run. We plot the super arm and group regret in Figure 2. First, notice that ACC-UCB incurs linear group regret as it does not take group thresholds into account. On the other hand, STCGP-UCB incurs very low group regret as it accounts for group thresholds. Additionally, the super arm regret of STCGP-UCB is less than that of ACC-UCB, indicating that even though STCGP-UCB is balancing between minimizing both group and super arm regret, it still outperforms ACC-UCB. This is likely due to the use of GPs, which allow for faster convergence to $\boldsymbol{f}$.

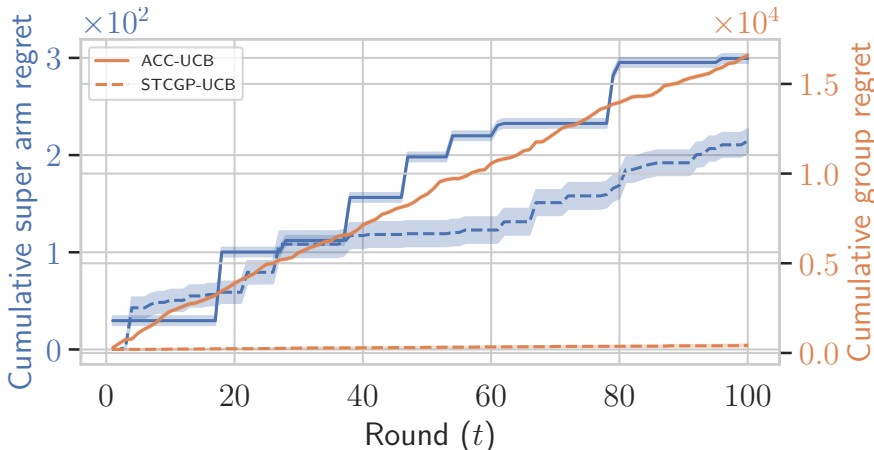

Figure 2: Cumulative super arm and group regret of ACC-UCB and STCGP-UCB. Super arm and group regret are represented by blue and orange, respectively. Moreover, ACC-UCB and STCGP-UCB are represented by solid and dashed lines, respectively. Shaded regions indicate $\pm$ std.

### 5.2 Caching-aware Movie Recommendation

### 5.2.1 Setup

We evaluate the proposed algorithm in a real-world dataset where our setup, similar to that of (Nika et al., 2021), is movie recommendation with content caching. We use the MovieLens 25M dataset with a slight modification (Harper & Konstan, 2015). This dataset consists of users, movies, and the ratings given to the movies by users. In order to motivate our definition of groups, we also add a randomly generated location for each user coming from a set of 10 different locations and define movie-user pairs (i.e., base arms) that have the same location as a group. Moreover, every movie has genre metadata coming from a group of 20 genres such as, but not limited to, action, adventure, and comedy. Every rating given to a movie by a user falls between 0.5 and 5.0, with increments of 0.5. In this dataset, we take into account only the ratings after 2015 and users who have rated at least 200 movies.

In each round $t$, we randomly choose $H_t$ movies from the available set of movies, where $H_t$ is Poisson distributed with a mean of 75. Next, we randomly select $P_t$ users who have reviewed the selected movies, where $P_t$ is Poisson distributed with a mean of 200. If a user $j$ has rated a movie $i$, then the context of this base arm (i.e., the pair of movie $i$ and user $j$) is given by $x_{i,j}$. This context is computed as $x_{i,j} = \langle \boldsymbol{u}_j, \boldsymbol{g}_i \rangle / 10$,

where $\boldsymbol{u}_j$ is the average of the genres of the movies that user $j$ rated, weighed by their rating, $\boldsymbol{g}_i$ is the genre vector of movie $i$, and 10 is a normalizing factor. [6]

We let movie-user pairs that are in the same location be groups and combinations of various movie-user pairs be super arms. We let the feasible set of super arms be any super arm with a size of $K = 20$. The expected base arm outcome function that is used for super arm reward computation, $f_1$, is given by $f_1(x_{i,j}) = x$ and the super arm reward is calculated as:

$$u(f_1(\boldsymbol{x}_{t,S})) = \sum_{j=1}^{P_t} \sum_{i=1}^{H_t} f_1(x_{i,j})$$

which is subject to $x_{i,j} \in \boldsymbol{x}_{t,S}$. This is motivated by the fact that when we are trying to maximize super arm reward, we are actually trying to pick the super arm whose base arms (i.e., movie-user pairs) are the ones in which the users' interests were aligned with the genre of the movies that were recommended to them.

Then, we define the expected base arm outcome function that is used for group reward computation, $f_2$, as $f_2(x_{i,j}) = 2/(1 + e^{-4x_{i,j}}) - 1$ and calculate the group reward according to the Dixit-Stiglitz model, as used in the simulations of (Chen et al., 2018), given by

$$v_G(f_2(\boldsymbol{x}_{G \cap S_t})) = \sum_{i=1}^{H_{t,l}} \left( \sum_{j=1}^{P_{t,l}^i} f_2(x_{i,j})^3 \right)^{1/3},$$

where $P_{t,l}^i$ is the number of users who have rated the $i$th movie and reside in the $l$th location, and $H_{t,l}$ is the number of movies that have been rated by at least one user residing in the $l$th location. Notice that this is a submodular function and the generality of our method enables us to deal with such functions. The movies of the base arms of the chosen super arm are cached, therefore, groups want to have their base arms to be contained in the selected super arm as much as possible, hence, the outcome function $f_2$ is a sigmoid-like function which gives a higher weight to the movies having a higher rating. On the other hand, the outcome function $f_1$ is the identity function and treats all the base arms fairly without favoring one over the other. Lastly, we get the random outcome used in our simulations by adding zero-mean Gaussian noise with standard deviation 0.05 to the expected outcomes.

### 5.2.2 Algorithms

We run the simulation using the same algorithms as the previous simulation, namely our algorithm with a sparse GP approximator, STCGP-UCB, and the non-GP C3-MAB state-of-the-art, ACC-UCB of (Nika et al., 2020).

**STCGP-UCB** In our simulation, we set the number of inducing points to $s = 5$. We also set $\delta = 0.05$, $\zeta = 0.5$, and use two Matern kernels with both lengthscale and variance set to 1.

**ACC-UCB**

We set $v_1 = 1, v_2 = 1, \rho = 0.5$, and $N = 2$, as given in Definition 1 of (Nika et al., 2020). The initial (root) context cell, $X_{0,1}$, is a line centered at $(0.5)$.

### 5.2.3 Results

We run the simulation for 200 rounds with twenty independent runs, averaging over the results of each run. We plot the cumulative super arm reward of each algorithm divided by that of the benchmark in Figure 3. As seen in the Figure, both algorithms managed to quickly learn which contexts are high-outcome, when it comes to the super arm reward. However, TCGP-UCB clearly learns much faster than ACC-UCB. In just $t = 26$ rounds, TCGP-UCB achieves an average cumulative reward of $0.80 \pm 0.11$, whereas it takes ACC-UCB more than twice as many rounds to reach the same reward; it reaches $0.80 \pm 0.06$ in round $t = 59$.

---

[6]We divide by 10 and not 20 to normalize the context because the maximum number of genres that a movie has in the dataset is 10.

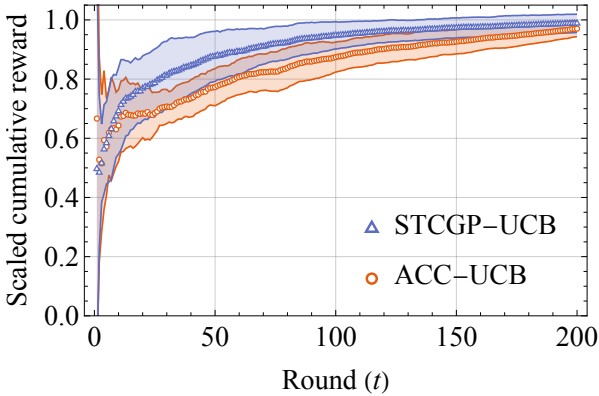
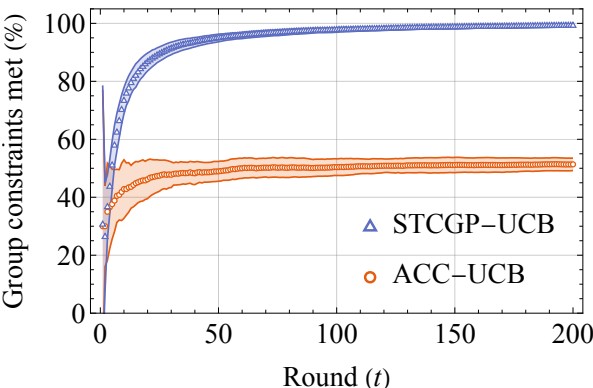

Figure 3: Cumulative super arm reward of each algorithm in each round divided by that of the benchmark, averaged over twenty independent runs. Error bands represent $\pm$ std.

Figure 4: Percentage of groups corresponding to picked super arms in each round that satisfied their constraints, averaged over twenty independent runs. Error bands represent $\pm$ std.

Moreover, the final cumulative reward of TCGP-UCB is also higher than that of ACC-UCB, albeit not by much: $0.988 \pm 0.031$ compared with $0.970 \pm 0.027$. Both the quickness of learning and higher final reward can likely be attributed to the fact that TCGP-UCB uses a GP, whose posterior update allows for faster learning than the adaptive discretization approach of ACC-UCB.

We also recorded the number of satisfied groups in each round for each algorithm. In other words, we kept track of the percentage of groups corresponding to the picked super arm in each round had their constraints met. We plot this result in Figure 4. First, we observe that the percentage of group constraints that TCGP-UCB satisfies quickly converges above 0.95 in 50 rounds. However, ACC-UCB struggles greatly with satisfying group constraints and manages to converge to around 50% of group constraints met. This discrepancy is simply because TCGP-UCB takes group constraints into account, while ACC-UCB does not.

## 6 Conclusion

We considered the C3-MAB problem with semi-bandit feedback, where in each round the agent has to play a feasible subset of the base arms in order to maximize the cumulative reward while satisfy group constraints. Our setup was motivated by privacy aware federated learning, where the goal is to optimize overall training result while satisfying clients' privacy requirements. Under the assumption that the expected base arm outcomes are drawn from a two-output GP and that the expected reward is Lipschitz continuous with respect to the expected base arm outcomes, we proposed TCGP-UCB, a double UCB algorithm that incurs $\tilde{O}(\sqrt{\lambda^*(K)KT\overline{\gamma}_T})$ regret in $T$ rounds. In experiments, we showed that sparse GPs can be used to speed up UCB computation, while simultaneously outperforming the state-of-the-art non-GP-based C3-MAB algorithm in a synthetic and real-world setup. Our comparisons also indicated that GPs can transfer knowledge among contexts better than partitioning the contexts into groups of similar contexts based on a similarity metric. An interesting future research direction involves investigating how dependencies between base arms can be used for more efficient exploration. Then, when the oracle selects base arms sequentially, it is possible to update the posterior variances of the not yet selected base arms by conditioning on the selected, but not yet observed, base arms.

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

## A   Table of Notation

We provide a table of notation so that the proof section can be easily followed.

| Symbol | Explanation |
|--------|-------------|
| $m$ | Variable to denote a base arm |
| $G$ | Variable to denote a group |
| $S$ | Variable to denote a super arm |
| $S_t$ | Variable to denote the chosen super arm at round $t$ |
| $\mathcal{M}_t$ | Set of base arms that are available at round $t$ |
| $M_t$ | Number of base arms that become available at round $t$ |
| $\mathcal{G}_t$ | Set of feasible groups in round $t$ |
| $\mathcal{G}_{t,\text{good}}$ | Set of groups whose expected rewards are above their thresholds |
| $\mathcal{S}_t$ | Set of feasible super arms in round $t$ |
| $\mathcal{S}'_t$ | Set of super arms whose corresponding groups satisfy their thresholds |
| $\hat{\mathcal{S}}'_t$ | Set of feasible super arms whose corresponding groups' reward indices are above their thresholds |
| $\mathcal{S}$ | Overall feasible set of super arms |
| $\mathcal{X}$ | Context set |
| $\mathcal{X}_t$ | Set of available contexts in round $t$ |
| $x_{t,m}$ | Context associated with base arm $m$ |
| $\boldsymbol{x}_{t,G}$ | Context vector of base arms in $G$ |
| $\boldsymbol{x}_{t,S}$ | Context vector of base arms in $S$ |
| $\boldsymbol{r}(x)$ | Random outcome of base arm with context $x$ |
| $\boldsymbol{f}(x)$ | Expected outcome of base arm with context $x$ |
| $\boldsymbol{\eta}$ | $\mathcal{N}(\boldsymbol{0}, \sigma^2 \boldsymbol{I})$ independent observation noise |
| $U(S, r_1(\boldsymbol{x}_{t,S}))$ | Random super arm reward |
| $u(S, f_1(\boldsymbol{x}_{t,S}))$ | Expected super arm reward function |
| $V_G(r_2(\boldsymbol{x}_{t,G\cap S_t}))$ | Random group reward |
| $v_G(f_2(\boldsymbol{x}_{t,G\cap S_t}))$ | Expected group reward function |
| $\gamma_{t,G}$ | Threshold for group $G$ at round $t$ |
| $\zeta$ | Trade-off parameter |
| $i_t(x_{t,m})$ | Index of base arm $m$ at round $t$ which is given to $\text{Oracle}_{\text{spr}}$ (reward index) |
| $i'_t(x_{t,m})$ | Index of base arm $m$ at round $t$ which is given to $\text{Oracle}_{\text{grp}}$ (satisfying index) |
| $\bar{\gamma}_T$ | Maximum information gain which is associated with the context arrivals $\mathcal{X}_1, \ldots, \mathcal{X}_T$ |
| $\gamma_N$ | Maximum information gain given $N$ base arm outcome observations |
| $K$ | Maximum possible number of base arms in a super arm |

When we state that $a \geq b$ with $a$ and $b$ being vectors, we mean that every component of $a$ is greater than or equal to the corresponding component of $b$. This holds for other comparison operators as well. Also, we omit $G$ from $v_G$ and $V_G$; and $S$ from $U(S, \ldots)$ and $u(S, \ldots)$ when it is obvious from the context. In the definition of $\mathcal{S}'_t$ and $\hat{\mathcal{S}}'_t$, the term corresponding groups means the groups that contain the base arms of the chosen super arm.

## B   Additional Experimental Results

### B.1   Changing Trade-off Parameter ($\zeta$)

In this simulation we showcase the behavior of our algorithm for changing trade-off parameter ($\zeta$) values.

### B.1.1 Setup

We use a synthetic setup where we generate the arm and group data needed for the simulation. Similar to the main paper simulations, in each round $t$, we first sample the number of groups, $|\mathcal{G}_t|$, from a Poisson distribution with mean 50. Then, for each group we generate the contexts of the base arms in that group, where the number of base arms is sampled from a Poisson distribution with mean 5. Each base arm has a two-dimensional (2D) context that is sampled uniformly from $[0, 1]^2$. Then, we sample the expected outcome of each base arm of each group from a GP with zero mean and two squared exponential kernels, each given by

$$k(x, x') = \exp\left(-\frac{1}{2l^2}\|x - x'\|\right),$$

where we set $l = 0.5$. Note that given that we run the simulation for $T = 100$ rounds, there will be an expected number of 25000 arms and to sample a GP function with that many points we will need to compute the Cholesky decomposition of a 625 million element matrix, which would be very resource-heavy. Instead, we first sample 6000 2D contexts from $[0, 1]^2$ and then sample the GP function at those points. Then, during our simulation, we sample each base arm's context $x$ and corresponding expected outcome $\boldsymbol{f}(x)$ from the generated sets. Finally, we set $\boldsymbol{r}(x) = \boldsymbol{f}(x) + \boldsymbol{\eta}$, where $\boldsymbol{\eta} \sim \mathcal{N}(\mathbf{0}, 0.1^2\boldsymbol{I}_2)$. We set the group reward, $v$, to be the variance of the outcomes in the group and we set the threshold to be the 80% percentile of the group rewards of all groups in all rounds of the simulation. We use a high percentile value to increase the difficulty of the group thresholding, so that minimizing super arm regret does not necessarily yield minimizing group regret. Finally, the super arm reward is the linear sum of the base arms.

### B.1.2 Results

We run our algorithm using five different values of $\zeta$, linearly spaced between 0.001 and 0.999. Figure 5 shows the final super arm and group regret of each $\zeta$ run (i.e., the cumulative regrets at the end of the simulation). First, notice a trade-off between super arm and group regret. As one increases, the other decreases. This is expected because to minimize group regret, groups with arms whose outcomes are spread out and have high variance must be picked, but to minimize super arm regret, groups with high outcomes must be picked. Second, as $\zeta$ increases, the super arm regret increases while the group regret decreases. This is because the variance of the indices given to the first oracle ($i'$), which determines which groups pass their thresholds, decreases as $\zeta$ approaches 1. Thus, the larger $\zeta$ is, the stricter the first oracle is. Conversely, the smaller $\zeta$ is the laxer the first oracle and the stricter the second oracle, which determines which super arm to play.

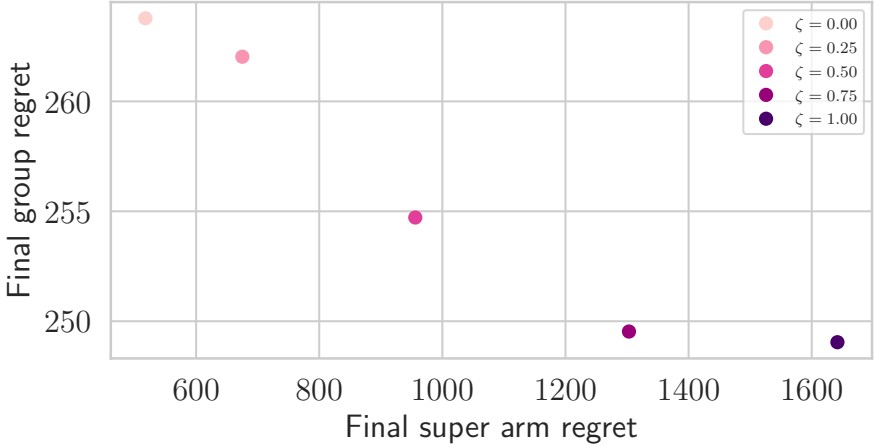

Figure 5: Final super arm and group regret for different trade-off parameter, $\zeta$, values

## C  Proofs

### C.1  Auxiliary Proofs

In this part, we will prove results that will be used in the proofs of the theorems in the paper.

Throughout this section, we take $j \in \{1, 2\}$. We denote by $\boldsymbol{r}_{j\llbracket t-1 \rrbracket}$ the vector of base arm outcome observations made until the beginning of round $t$, where

$$\boldsymbol{r}_{j\llbracket t-1 \rrbracket} = [r_j^T(\boldsymbol{x}_{1,S_1}), \dots, r_j^T(\boldsymbol{x}_{t-1,S_{t-1}})]^T.$$

For any $t \geq 1$, the posterior distribution of $f_j(x)$ given the observation vector $\boldsymbol{r}_{j\llbracket t-1 \rrbracket}$ is $\mathcal{N}(\mu_{j\llbracket t-1 \rrbracket}(x), (\sigma_{j\llbracket t-1 \rrbracket}(x))^2)$, for any $x \in \mathcal{X}_t$. In our analysis, we will resort to the following Gaussian tail bound

$$\mathbb{P}\left(|f_j(x) - \mu_{j\llbracket t-1 \rrbracket}(x)| > (\sqrt{\beta_t})\sigma_{j\llbracket t-1 \rrbracket}(x) \Big| \boldsymbol{r}_{j\llbracket t-1 \rrbracket}\right) \leq 2\exp\left(\frac{-\beta_t}{2}\right) \text{ for } \beta_t \geq 0 . \tag{3}$$

The following lemma shows that the base arm indices upper bound the expected outcomes with high probability.

**Lemma 1.** *(Lemma 1 of (Nika et al., 2021)) Fix $\delta \in (0,1)$, and set $\beta_t := 2\log\left(M\pi^2 t^2/3\delta\right)$. Let $\mathcal{F}_j := \{\forall t \geq 1, \forall x \in \mathcal{X}_t : |f_j(x) - \mu_{j\llbracket t-1 \rrbracket}(x)| \leq (\sqrt{\beta_t})\sigma_{j\llbracket t-1 \rrbracket}(x)\}$. We have $\mathbb{P}(\mathcal{F}_j) \geq 1 - \delta$.*

Now, we state that the modified indices upper bound the expected base arm outcomes with high probability under the events $\mathcal{F}_j$.

**Lemma 2.** *The following arguments hold under $\mathcal{F}_1 \cap \mathcal{F}_2$.*

$$i_t(x) \geq f_1(x), \forall t \geq 1, \ \forall x \in \mathcal{X}_t$$
$$i'_t(x) \geq f_2(x), \forall t \geq 1, \ \forall x \in \mathcal{X}_t . \tag{4}$$

*Proof.* Fix $t \geq 1$ and $x \in \mathcal{X}_t$. Under event $\mathcal{F}_1$, we have the following chain of inequalities

$$f_1(x) - \mu_{1\llbracket t-1 \rrbracket}(x) \leq (\sqrt{\beta_t})\sigma_{1\llbracket t-1 \rrbracket}(x)$$
$$f_1(x) - \mu_{1\llbracket t-1 \rrbracket}(x) \leq \frac{1}{1-\zeta}(\sqrt{\beta_t})\sigma_{1\llbracket t-1 \rrbracket}(x) \tag{5}$$
$$f_1(x) \leq \mu_{1\llbracket t-1 \rrbracket}(x) + \frac{1}{1-\zeta}(\sqrt{\beta_t})\sigma_{1\llbracket t-1 \rrbracket}(x) = i_t(x) .$$

Proceeding in the same fashion for event $\mathcal{F}_2$ , we obtain

$$f_2(x) - \mu_{2\llbracket t-1 \rrbracket}(x) \leq \frac{1}{\zeta}(\sqrt{\beta_t})\sigma_{2\llbracket t-1 \rrbracket}(x) \tag{6}$$
$$f_2(x) \leq \mu_{2\llbracket t-1 \rrbracket}(x) + \frac{1}{\zeta}(\sqrt{\beta_t})\sigma_{2\llbracket t-1 \rrbracket}(x) = i'_t(x) ,$$

where equation 5 and equation 6 follow from the fact that $\frac{1}{1-\zeta} \geq 1$ and $\frac{1}{\zeta} \geq 1$ when $\zeta \in [0,1]$.

$\qquad\square$

Next, we show that the set of feasible super arms whose corresponding groups satisfy their thresholds is a subset of the set of feasible super arms whose groups' reward indices are above their thresholds. We later use this result in Lemma 5.

**Lemma 3.** *Fix $\delta \in (0,1)$. The following argument holds under the event $\mathcal{F}_2$ when the group reward function $v$ satisfies the monotonicity assumption given in Assumption 1.*

$$\mathcal{S}'_t \subseteq \hat{\mathcal{S}}'_t, \ \forall t \geq 1.$$

*Proof.* For any $S_t$ and for all $G$ we have:

$$S_t \in \mathcal{S}'_t \iff v(f_2(\boldsymbol{x}_{t,G\cap S_t})) \geq \gamma_{t,G} \tag{7}$$
$$\iff v(f_2(\boldsymbol{x}_{t,G\cap S_t})) - \gamma_{t,G} \geq 0$$
$$\implies v(i'_t(\boldsymbol{x}_{t,G\cap S_t})) - \gamma_{t,G} \geq 0 \tag{8}$$
$$\iff S_t \in \hat{\mathcal{S}}'_t \tag{9}$$

where equation 7 follows from the definition of $\mathcal{S}'_t$, equation 8 follows from the inequality that $v(i'_t(\boldsymbol{x}_{t,G\cap S_t})) \geq v(f_2(\boldsymbol{x}_{t,G\cap S_t}))$. Since $i'_t(\boldsymbol{x}_{t,G\cap S_t}) \geq f_2(\boldsymbol{x}_{t,G\cap S_t})$ under the event $\mathcal{F}_2$ and $v$ is monotone by assumption, this inequality is valid. Lastly, equation 9 follows from the fact that $\text{Oracle}_{\text{grp}}$ is an exact oracle and will return the groups where $v(i'_t(\boldsymbol{x}_{t,G\cap S_t})) > \gamma_{t,G}$. As this reasoning is true for any $S_t$, we indicate that $\mathcal{S}'_t \subseteq \hat{\mathcal{S}}'_t$.

Detail:

$$\mathcal{S}'_t := \{S \in \mathcal{S}_t : \forall (G \in \mathcal{G}_t), v_G(f_2(\boldsymbol{x}_{t,G\cap S})) \geq \gamma_{t,G}\}.$$
$$\hat{\mathcal{S}}'_t := \{S \in \mathcal{S}_t : \forall G \in \mathcal{G}_t, \ S \cap G \in \mathcal{G}_{t,\text{good}}\}$$
$$\mathcal{G}_{t,\text{good}} := \{G' \subseteq G \mid G \in \mathcal{G}_t \text{ and } v_G(i'_t(\boldsymbol{x}_{t,G'})) \geq \gamma_{t,G'}\}$$
$$i'_t(x_{t,m}) := \mu_{2[\![t-1]\!]}(x_{t,m}) + \frac{1}{\zeta}(\sqrt{\beta_t})\sigma_{2[\![t-1]\!]}(x_{t,m}).$$

$\square$

Next, we upper bound the group regret in terms of the posterior variances of base arms. Note that group regret is incurred when a selected group's expected reward is below its threshold whereas its index is above. Therefore, whenever a group $G$ incurs group regret in round $t$, then $v(f_2(\boldsymbol{x}_{t,G\cap S_t})) < \gamma_{t,G} < v(i'_t(\boldsymbol{x}_{t,G\cap S_t}))$ must happen. This observation plays a key role in the analysis of the next lemma. Moreover, we use the notation $\tilde{x}_{t,k}$ to denote the context of the $k$th base arm in $G \cap S_t$ at round $t$ for convenience, unless otherwise stated.

**Lemma 4.** *Fix $t \geq 1$, and consider $G \in \mathcal{G}_t$. The following argument holds under the event $\mathcal{F}_2$:*

$$[\gamma_{t,G} - v(f_2(\boldsymbol{x}_{t,G\cap S_t}))]_+ \leq \left(\frac{\zeta+1}{\zeta}\right) B\sqrt{\beta_t} \sum_{k=1}^{|G\cap S_t|} |\sigma_{2[\![t-1]\!]}(\tilde{x}_{t,k})| . \tag{10}$$

*Proof.* $[\gamma_{t,G} - v(f_2(\boldsymbol{x}_{t,G\cap S_t}))]_+ > 0$ implies that $v(f_2(\boldsymbol{x}_{t,G\cap S_t})) < \gamma_{t,G}$ and $v(i'_t(\boldsymbol{x}_{t,G\cap S_t})) \geq \gamma_{t,G}$. Therefore, whenever $G$ incurs group regret it holds that $v(f_2(\boldsymbol{x}_{t,G\cap S_t})) < \gamma_{t,G} \leq v(i'_t(\boldsymbol{x}_{t,G\cap S_t}))$.

$$0 < \gamma_{t,G} - v(f_2(\boldsymbol{x}_{t,G\cap S_t})) < v(i'_t(\boldsymbol{x}_{t,G\cap S_t})) - v(f_2(\boldsymbol{x}_{t,G\cap S_t}))$$
$$0 < [\gamma_{t,G} - v(f_2(\boldsymbol{x}_{t,G\cap S_t}))]_+ < v(i'_t(\boldsymbol{x}_{t,G\cap S_t})) - v(f_2(\boldsymbol{x}_{t,G\cap S_t}))$$
$$\leq B_{G\cap S_t} \sum_{k=1}^{|G\cap S_t|} |i'_t(\tilde{x}_{t,k}) - f_2(\tilde{x}_{t,k})| \tag{11}$$
$$\leq B_{G\cap S_t} \sum_{k=1}^{|G\cap S_t|} |\mu_{2[\![t-1]\!]}(\tilde{x}_{t,k}) - f_2(\tilde{x}_{t,k})| + B_{G\cap S_t} \sum_{k=1}^{|G\cap S_t|} \left|\frac{1}{\zeta}\sqrt{\beta_t}\sigma_{2[\![t-1]\!]}(\tilde{x}_{t,k})\right| \tag{12}$$
$$\leq \left(\frac{\zeta+1}{\zeta}\right) B(\sqrt{\beta_t}) \sum_{k=1}^{|G\cap S_t|} |\sigma_{2[\![t-1]\!]}(\tilde{x}_{t,k})| , \tag{13}$$

where equation 11 follows from the Lipschitz continuity of $v$; equation 12 follows from the definition of index and the triangle inequality; for equation 13 we use Lemma 1 and the definition of $B$.

$\square$

Next, we upper bound the gap of a selected super arm in round $t$ (aka simple regret) in terms of the posterior variances of base arms. Hereafter, we use the notation $\overline{x}_{t,k}$ to denote the context $x_{t,s_{t,k}}$ of the $k$th selected base arm $s_{t,k}$ at round $t$ for convenience, unless otherwise stated. Note that $S_t^* \in \arg\max_{S \in \mathcal{S}_t'} u(f_1(\boldsymbol{x}_{t,S}))$ is the optimal super arm in round $t$.

**Lemma 5.** *Given round $t \geq 1$, let $S_t^* = \{s_{t,1}^*, \ldots, s_{t,|S_t^*|}^*\}$ denote the optimal super arm in round $t$. Then, the following argument holds under the event $\mathcal{F}_1$:*

$$\alpha \cdot u(f_1(\boldsymbol{x}_{t,S_t^*})) - u(f_1(\boldsymbol{x}_{t,S_t})) \leq \left(\frac{2-\zeta}{1-\zeta}\right) B' \sqrt{\beta_t} \sum_{k=1}^{|S_t|} |\sigma_{1\llbracket t-1 \rrbracket}(\overline{x}_{t,k})| \tag{14}$$

*Proof.* We define $H_t = \arg\max_{S \in \hat{\mathcal{S}}_t'} u(i_t(\boldsymbol{x}_{t,S}))$. Given that event $\mathcal{F}_1$ holds, we have:

$$\alpha \cdot u(f_1(\boldsymbol{x}_{t,S_t^*})) - u(f_1(\boldsymbol{x}_{t,S_t})) \leq \alpha \cdot u(i_t(\boldsymbol{x}_{t,S_t^*})) - u(f_1(\boldsymbol{x}_{t,S_t})) \tag{15}$$

$$\leq \alpha \cdot u(i_t(\boldsymbol{x}_{t,H_t})) - u(f_1(\boldsymbol{x}_{t,S_t})) \tag{16}$$

$$\leq u(i_t(\boldsymbol{x}_{t,S_t})) - u(f_1(\boldsymbol{x}_{t,S_t})) \tag{17}$$

$$\leq B' \sum_{k=1}^{|S_t|} |i_t(\overline{x}_{t,k}) - f_1(\overline{x}_{t,k})| \tag{18}$$

$$\leq B' \sum_{k=1}^{|S_t|} |\mu_{1\llbracket t-1 \rrbracket}(\overline{x}_{t,k}) - f_1(\overline{x}_{t,k})| + B' \sum_{k=1}^{|S_t|} \left|\frac{1}{1-\zeta}(\sqrt{\beta_t})\sigma_{1\llbracket t-1 \rrbracket}(\overline{x}_{t,k})\right| \tag{19}$$

$$\leq \left(\frac{2-\zeta}{1-\zeta}\right) B' \sqrt{\beta_t} \sum_{k=1}^{|S_t|} |\sigma_{1\llbracket t-1 \rrbracket}(\overline{x}_{t,k})| , \tag{20}$$

where equation 15 follows from monotonicity of $u$ and the fact that $f_1(x_{t,s_{t,k}^*}) \leq i_t(x_{t,s_{t,k}^*})$, for $k \leq |S_t^*|$ (Lemma 2); equation 16 follows from the definition of $H_t$ and the fact that $\mathcal{S}_t' \subseteq \hat{\mathcal{S}}_t'$ on event $\mathcal{F}_2$ (Lemma 3), in other words, $\max_{S \in \hat{\mathcal{S}}_t'} u(i_t(\boldsymbol{x}_{t,S})) \geq \max_{S \in \mathcal{S}_t'} u(i_t(\boldsymbol{x}_{t,S}))$ since $\mathcal{S}_t' \subseteq \hat{\mathcal{S}}_t'$; equation 17 holds since $S_t$ is the super arm chosen by the $\alpha$-approximation oracle; equation 18 follows from the Lipschitz continuity of $u$; equation 19 follows from the definition of index and the triangle inequality; for equation 20 we use Lemma 1.

Before proving our theorems, we prove our last lemma which enables us to have our regrets bounds in terms of the information gain. We note that both of our oracles are deterministic and our algorithm doesn't give any extra randomization.

**Lemma 6.** *(Lemma 3 of (Nika et al., 2021)) Let $\boldsymbol{z}_t := \boldsymbol{x}_{t,S_t}$ be the vector of selected contexts at time $t \geq 1$. Given $T \geq 1$, we have:*

$$I_j\left(r_j(\boldsymbol{z}_{[T]}); f_j(\boldsymbol{z}_{[T]})\right) \geq \frac{1}{2(\sigma^{-2}\lambda^*(K)+1)} \sum_{t=1}^{T} \sum_{k=1}^{|S_t|} \sigma^{-2}\sigma_{j\llbracket t-1 \rrbracket}^2(\overline{x}_{t,k}) ,$$

*where $\boldsymbol{z}_{[T]} = [\boldsymbol{z}_1, \ldots, \boldsymbol{z}_T]^T$ is the vector of all selected contexts until round $T$ and $\lambda^*$ is the maximum eigenvalue of matrix $(\Sigma_{\llbracket t-1 \rrbracket}(\boldsymbol{z}_{[T]}))_{t=1}^{T}$.*

## C.2 Proof of Theorem 1

From Lemma 4 we have:

$$R_g(T) = \sum_{t=1}^{T} \sum_{G \in \tilde{\mathcal{G}}_t} [\gamma_{t,G} - v(f_2(\boldsymbol{x}_{t,G \cap S_t}))]_+$$

$$\leq \left(\frac{\zeta+1}{\zeta}\right) B\sqrt{\beta_T} \sum_{t=1}^{T} \sum_{G \in \mathcal{G}_t} \sum_{k=1}^{|G \cap S_t|} |\sigma_{2[\![t-1]\!]}(\tilde{x}_{t,k})|$$

$$\leq \left(\frac{\zeta+1}{\zeta}\right) B\sqrt{\beta_T} \sum_{t=1}^{T} \sum_{k=1}^{|S_t|} |\sigma_{2[\![t-1]\!]}(\overline{x}_{t,k})| \ , \tag{21}$$

using the fact that $\sqrt{\beta_t}$ is monotonically increasing in $t$. Also, we changed the notation of $\tilde{x}_{t,k}$ with $\overline{x}_{t,k}$ as we are summing through all the base arms in $S_t$ in equation 21. We have:

$$R_g^2(T) \leq \left(\frac{\zeta+1}{\zeta}\right)^2 B^2 \beta_T \left(\sum_{t=1}^{T} \sum_{k=1}^{|S_t|} |\sigma_{2[\![t-1]\!]}(\overline{x}_{t,k})|\right)^2 \tag{22}$$

$$\leq \left(\frac{\zeta+1}{\zeta}\right)^2 B^2 \beta_T T \sum_{t=1}^{T} \left(\sum_{k=1}^{|S_t|} |\sigma_{2[\![t-1]\!]}(\overline{x}_{t,k})|\right)^2 \tag{23}$$

$$\leq \left(\frac{\zeta+1}{\zeta}\right)^2 B^2 \beta_T T \sum_{t=1}^{T} |S_t| \sum_{k=1}^{|S_t|} \left(\sigma_{2[\![t-1]\!]}(\overline{x}_{t,k})\right)^2 \tag{24}$$

$$\leq \left(\frac{\zeta+1}{\zeta}\right)^2 B^2 \beta_T T K \sum_{t=1}^{T} \sum_{k=1}^{|S_t|} \left(\sigma_{2[\![t-1]\!]}(\overline{x}_{t,k})\right)^2$$

$$\leq \left(\frac{\zeta+1}{\zeta}\right)^2 B^2 \beta_T T K \sigma^2 \sum_{t=1}^{T} \sum_{k=1}^{|S_t|} \sigma^{-2} \left(\sigma_{2[\![t-1]\!]}(\overline{x}_{t,k})\right)^2 \tag{25}$$

$$\leq 2(\sigma^{-2}\lambda^*(K) + 1) \left(\frac{\zeta+1}{\zeta}\right)^2 B^2 \beta_T T K \sigma^2 I\left(r_2(\boldsymbol{z}_{[T]}); f_2(\boldsymbol{z}_{[T]})\right) \tag{26}$$

$$\leq C_1(K) K \beta_T T \overline{\gamma_2}_T \ , \tag{27}$$

where for equation 23 and equation 24 we have used the Cauchy-Schwarz inequality twice; in equation 25 we just multiply by $\sigma^2$ and $\sigma^{-2}$; equation 26 follows from Lemma 6 and for equation 27 we use the definition of $\overline{\gamma_2}_T$. Taking the square root of both sides we obtain our desired result.

From Lemma 5 we have:

$$R_s(T) = \alpha \sum_{t=1}^{T} \text{opt}(f_t) - \sum_{t=1}^{T} u(f_1(\boldsymbol{x}_{t,S_t}))$$

$$\leq \left(\frac{2-\zeta}{1-\zeta}\right) B'\sqrt{\beta_T} \sum_{t=1}^{T} \sum_{k=1}^{|S_t|} |\sigma_{1[\![t-1]\!]}(\overline{x}_{t,k})| \ , \tag{28}$$

using the fact that $\sqrt{\beta_t}$ is monotonically increasing in $t$. We have:

$$R_s^2(T) \leq \left(\frac{2-\zeta}{1-\zeta}\right)^2 (B')^2 \beta_T \left(\sum_{t=1}^{T} \sum_{k=1}^{|S_t|} |\sigma_{1[\![t-1]\!]}(\overline{x}_{t,k})|\right)^2 \tag{29}$$

The middle steps are the same as equation 23-equation 26 except that we have $\left(\frac{2-\zeta}{1-\zeta}\right)^2 (B')^2 \beta_T$ instead of $\left(\frac{\zeta+1}{\zeta}\right)^2 B^2 \beta_T$ as the constant multiplier. Also, we use $\sigma_1$ instead of $\sigma_2$. Hence, the last steps are modified as:

$$R_s^2(T) \leq 2(\sigma^{-2}\lambda^*(K)+1)\left(\frac{2-\zeta}{1-\zeta}\right)^2 (B')^2 \beta_T T K \sigma^2 I\left(r_1(\boldsymbol{z}_{[T]}); f_1(\boldsymbol{z}_{[T]})\right)$$
$$\leq C_2(K)K\beta_T T\overline{\gamma_1}_T \ ,$$

Taking the square root of both sides we obtain our desired result. Finally, in order to obtain the total regret we use:

$$R(T) = \zeta R_g(T) + (1-\zeta)R_s(T)$$
$$\leq \left(\zeta\sqrt{C_1(K)} + (1-\zeta)\sqrt{C_2(K)}\right)\sqrt{\beta_T KT\overline{\gamma}_T}$$

where $\overline{\gamma}_T = \max\{\overline{\gamma_1}_T, \overline{\gamma_2}_T\}$. In order to eliminate the $\zeta$ dependence we modify this expression as follows:

$$R(T) \leq \left(\zeta\sqrt{2B^2(\frac{\zeta+1}{\zeta})^2(\lambda^*(K)+\sigma^2)} + (1-\zeta)\sqrt{2(B')^2(\frac{2-\zeta}{1-\zeta})^2(\lambda^*(K)+\sigma^2)}\right)\sqrt{\beta_T KT\overline{\gamma}_T} \quad (30)$$

$$= \left(\frac{\zeta}{|\zeta|}\sqrt{2B^2(\zeta+1)^2(\lambda^*(K)+\sigma^2)} + \frac{1-\zeta}{|1-\zeta|}\sqrt{2(B')^2(2-\zeta)^2(\lambda^*(K)+\sigma^2)}\right)\sqrt{\beta_T KT\overline{\gamma}_T}$$

$$= \left(\sqrt{2B^2(\zeta+1)^2(\lambda^*(K)+\sigma^2)} + \sqrt{2(B')^2(2-\zeta)^2(\lambda^*(K)+\sigma^2)}\right)\sqrt{\beta_T KT\overline{\gamma}_T} \quad (31)$$

$$= \sqrt{2(\lambda^*(K)+\sigma^2)}\left(|B||\zeta+1| + |B'||2-\zeta|\right)\sqrt{\beta_T KT\overline{\gamma}_T}$$

$$= \sqrt{2(\lambda^*(K)+\sigma^2)}\left(B(\zeta+1) + B'(2-\zeta)\right)\sqrt{\beta_T KT\overline{\gamma}_T} \quad (32)$$

$$\leq \sqrt{2(\lambda^*(K)+\sigma^2)}\left(2B + 2B'\right)\sqrt{\beta_T KT\overline{\gamma}_T} \quad (33)$$

$$= \sqrt{C(K)\beta_T KT\overline{\gamma}_T} \ .$$

where equation 30 follows from writing the expressions of $C_1$ and $C_2$ to the required places; equation 31 follows from $\zeta \in [0,1]$; equation 32 follows from the assumptions that $B > 0$ and $B' > 0$ and also from $\zeta \in [0,1]$ and equation 33 comes from writing the maximizing $\zeta$ values in $\zeta + 1$ and $2 - \zeta$.

### C.3  Proof of Theorem 2

The proof is the same as that of Theorem 2 of (Nika et al., 2021).

### C.4  Proof of Theorem 3

We first state an auxiliary fact that will be used in the proof (for a proof see (Williams & Vivarelli, 2000)).

**Fact 1.** *The predictive variance of a given context is monotonically non-increasing (i.e., given $x \in \mathcal{X}$ and the vector of selected samples $[x_1, \ldots, x_{N+1}]$, we have $\sigma_{N+1}^2(x) \leq \sigma_N^2(x)$, for any $N \geq 1$).*

The proof of Theorem 3 easily follows from Theorem 2 and this fact.

### C.5  Proof of Corollary 1

This is a direct application of the explicit bounds on $\gamma_T$ given in Theorem 5 of (Srinivas et al., 2012) to the bound we obtained in Theorem 3. For the Matérn kernel, we have used the tighter bounds from (Vakili et al., 2020).

