# OpenReview forum: "Contextual Combinatorial Multi-output GP Bandits with Group Constraints"
_TMLR — Accepted by TMLR_

### Review · Reviewer_98o9 · 2023-02-06

**Summary Of Contributions:**

The paper studies a specific instance of the combinatorial mab problem in which the distribution of the reward and the constraint we have on a subset of the arms follows a GP distribution. The authors propose an algorithm based on the UCB approach to minimize a specific definition of regret that takes into account the suboptimality of the selected arms and the violation of the abovementioned constraints. The authors propose a theoretical analysis of the proposed algorithm and a brief experimental analysis of the proposed solution.


**Audience:**

Yes

**Broader Impact Concerns:**

I do not think that the paper has ethical implications.

**Claims And Evidence:**

Yes

**Requested Changes:**

Regret definition: can you extend the framework to consider the number of times the constraint is violated?
From the definition of the problem you provided in section 2.8, is seems that a regret formulation in which the number of violations is considered would be more appropriate.

Information gain: It would help to evaluate the complexity of the model having an explicit relationship between the two information gains.
Moreover, while it is well studied the behaviour of the information gain on standard GP, I would like to have more information and comments on this new definition of information gain. For instance, how does it scales with standard linear and gaussian kernels

Parameter \zeta: it is not clear what this parameter is in the setting you are describing. Indeed, the two regrets are not comparable, therefore considering them in a linearly combined way does not provide any information on the goodness of the proposed method.

Comparison: you should compare this approach with other gaussian or GP-based solutions for CMAB. For instance:
- Wang, Siwei, and Wei Chen. "Thompson sampling for combinatorial semi-bandits." International Conference on Machine Learning. PMLR, 2018.
- Nuara, Alessandro, et al. "A combinatorial-bandit algorithm for the online joint bid/budget optimization of pay-per-click advertising campaigns." Proceedings of the AAAI Conference on Artificial Intelligence. Vol. 32. No. 1. 2018.

The experimental section is really poor. The experiments are limited to a specific synthetically generated setting and are not providing enough evidence comparing what is proposed here with state-of-the-art. The same comment hold also for the experiments provided in the supplemental material.

Minor changes:
We consider the scenario when -> We consider the scenario in which
Section 1.1 please explain the difference between \bar{\gamma}_T and \gamma_T
In accord with our goal -> In accordance with our goal,
reduces the problem to a combinatorial version of satisfying -> not clear


**Strengths And Weaknesses:**

Pros:
I think that the paper is motivated by a specific applicative example (federated learning with privacy constraints)


Cons:
The problem definition is somehow not coherent with the target application.

I think the newly defined regret concept can be further investigated to fit better the considered applicative case.

The dependence of the information gained w.r.t. the characterizing quantity of the setting should be detailed.

The experimental section is not fully convincing since it has been based on a single synthetic experiment whose parameters are not likely to represent a real-world case.

---

> ### Author Response · Authors · 2023-04-03
> **Response to reviewer 98o9**
>
> We thank the reviewer for the comments and questions. We addressed them below as well as in the revised manuscript.
>
> **Comment 1:**
> *Regret definition: can you extend the framework to consider the number of times the constraint is violated? From the definition of the problem you provided in section 2.8, is seems that a regret formulation in which the number of violations is considered would be more appropriate.*
>
> **Response:**
> It is indeed possible to formulate the group regret to consider the number of times the constraint is violated. Notice that our proposed algorithm, TCGP-UCB, would work just as well in this setting because the group oracle identifies good subgroups, which all satisfy their threshold (see section 2.4.1). However, we made the design choice of defining the group regret by using the difference between the threshold and the expected reward of a group.
>
> **Comment 2:**
> *Information gain: It would help to evaluate the complexity of the model having an explicit relationship between the two information gains. Moreover, while it is well studied the behaviour of the information gain on standard GP, I would like to have more information and comments on this new definition of information gain. For instance, how does it scales with standard linear and Gaussian kernels.*
>
> **Response:**
> We added lower and upper bounds on $\bar{\gamma}_T$, given in Theorem 2. We also added explicit kernel-dependent bounds for the linear, RBF, and Matern kernels, given in Corollary 1.
>
> **Comment 3:**
> *Parameter $\zeta$: it is not clear what this parameter is in the setting you are describing. Indeed, the two regrets are not comparable, therefore considering them in a linearly combined way does not provide any information on the goodness of the proposed method.*
>
> **Response:**
> $\zeta$ is a trade-off parameter which indicates how much the clients in the federated learning setup value their privacy. If the clients on average value their privacy a lot the $\zeta$ value will be closer to 1 and we will be prioritizing to minimize group regret. A $\zeta$ value closer to 1 comes at the cost of a lower amount of information retrieved by the server hence a higher super arm regret. Our combination of the two regrets using $\zeta$ was inspired by the scalarization approach (Boyd & Vandenberghe, 2004) that is often used in multi-objective problems. For a more detailed explanation please take a look at section 2.3 (Regret) in our revised version of the paper.
>
> **Comment 4:**
> *Comparison: you should compare this approach with other gaussian or GP-based solutions for CMAB.*
>
> **Response:**
> The combinatorial multi-armed bandit problem was first proposed and solved by (Chen et al., 2013). This problem has been thoroughly investigated by using upper confidence bound (UCB) (Chen et al., 2013; 2016b) and thompson sampling (TS) algorithms (Wang & Chen, 2018). While we also use a UCB type algorithm, we have groups and group constraints and we operate in a contextual combinatorial setting instead of a combinatorial one. In terms of the regret bounds, (Chen et al., 2013), and (Chen et al., 2016b) incur $O(\log T)$ regret whereas we incur $O(\sqrt{T})$ regret. (Wang & Chen, 2018) also incurs $O(\log T)$ but it is harder to compare with our bounds since their regret is gap dependent as well.
>
> In the combinatorial MAB setting of (Nuara et al., 2018) it has been shown that discretization works in small action spaces and Gaussian Processes are used to estimate the model.
>
> **Comment 5:**
> *The experiments are limited to a specific synthetically generated setting and are not providing enough evidence comparing what is proposed here with state-of-the-art.*
>
> **Response:**
> We performed new simulations on a caching-aware content recommendation setup using the MovieLens 25M dataset, a real-world dataset. We show that in this new simulation, our approach not only performs equally or better than the non-GP state-of-the-art, but that it does so while satisfying group constraints. The new simulation is given in Section 5.2 of the revised manuscript.
>
> **Comment 6:**
> *Minor changes: We consider the scenario when -> We consider the scenario in which Section 1.1 please explain the difference between $\bar{\gamma}_T$ and $\gamma_T$ In accord with our goal -> In accordance with our goal, reduces the problem to a combinatorial version of satisfying -> not clear*
>
> **Response:**
> We have made the listed minor changes. $\gamma_T$ is the classical notion of maximum information gain defined over all $N$-tuple of contexts over the context set $\mathcal{X}$, whereas, we have defined $\overline{\gamma}_T$ to be the maximum information gain defined over the contexts of the base arms from round 1 up to $T$, reducing the size of the context space that we take into account while computing the information gain. $\zeta=1$ reduces our problem to a thresholding/satisfying problem in a combinatorial setup, meaning the input space is the set of subsets of base arms (namely groups).

---

### Review · Reviewer_9crF · 2023-02-17

**Summary Of Contributions:**

This paper studies the problem of regret minimization of conspiratorial contextual bandits with groups and changing actions in the federated bandit scenario. The paper proposes a new algorithm TCGP-UCB for this new bandit problem and shows a regret upper bound \tilde{O}(\sqrt{KT\gamma}), where K is the number of arms, T is the time horizon, and \gamma is a parameter determined by the bandit bandit environment. The paper also conducts some numerical simulations and show improvements


**Audience:**

Yes

**Claims And Evidence:**

Yes

**Requested Changes:**

1. Make more justifications about the setup of the paper, including how it can be applied in practical use cases.
2. Show that the theoretical upper bound of the algorithm in this paper is good enough by i) making it clear how the \gamma term would change with the instance and why it is necessary; and ii) show lower bound results of the problem.

**Strengths And Weaknesses:**

Overall, this paper needs to show novelties or large contributions in one of the following two aspects.

    This paper's setup is not justified with reasonable real-world applications. There are too many additional requirements attached to the setup, which may limit the practical applications of the algorithm in the real world.
    This paper does not show high-quality theoretical results. To be more specific, the upper bounds of the algorithms have terms such as \gamma which is related to the maximal information gain. However, there is no clear statement about i) whether this term is necessary which can be shown by lower bound results but not included in this paper; and ii) How the \gamma term may increase with the number of actions, or other conditions of the environment.

In conclusion, this paper studies a new problem whose real-world applications/practical needs are not well illustrated. Besides, the theoretical results in this paper have many areas to improve. Thus, I tend to vote a major revision.

---

> ### Author Response · Authors · 2023-04-03
> **Response to Reviewer 9crF**
>
>
> We thank the reviewer for the comments and questions. We addressed them below as well as in the revised manuscript.
>
> **Comment 1:**
> *This paper's setup is not justified with reasonable real-world applications. There are too many additional requirements attached to the setup, which may limit the practical applications of the algorithm in the real world.
> Make more justifications about the setup of the paper, including how it can be applied in practical use cases.*
>
> **Response:**
> We have provided another application of our problem to a movie recommendation and content caching setup in our revised manuscript. The introductory explanation in Section 2.8 is given below but we would kindly like to encourage the reviewers to take a look at Section 5.2 for a more detailed explanation.
>
> Another motivation for our setup is movie recommendation with content caching. In this formulation, base arms can be movie-user pairs and context can represent the alignment between the users' interests and the movies' contents, which is captured by the rating. Groups can be taken to be the users that are in the same location and super arms can again be the combination of various base arms. When a super arm is selected, movies of the base arms that are contained in this super arm are cached in their respective locations, hence, maximizing super arm reward is related to caching content that will be watched and liked by people. In this problem, we want to identify which movies to recommend and cache and in which locations we should cache these movies. Hence, we want to maximize the super arm reward which is a function of the ratings of the movies that we recommend to the users, in order to cache these movies since they are rated highly. But we also want to ensure that the movies that we cached at a certain location are well-liked on average by the users at that location since this is the main goal of caching. This corresponds to maximizing group reward. We give a more detailed explanation with mathematical notation for these two setups in Section 5.
>
> **Comment 2:**
> *This paper does not show high-quality theoretical results. To be more specific, the upper bounds of the algorithms have terms such as $\gamma$ which is related to the maximal information gain. However, there is no clear statement about i) whether this term is necessary which can be shown by lower bound results but not included in this paper; and ii) How the $\gamma$ term may increase with the number of actions, or other conditions of the environment.
> Show that the theoretical upper bound of the algorithm in this paper is good enough by i) making it clear how the $\gamma$ term would change with the instance and why it is necessary; and ii) show lower bound results of the problem.*
>
> **Response:**
> We have provided a lower bound on the $\bar{\gamma}_T$ term that relates it to the classical notion of information gain in Theorem 2, which is repeated here for convenience.
>
>    Letting $\overline{T}=\sum^T_{t=1}|S_t|$ be the sum of cardinalities of selected actions up to time $T$, we have that
>
> $$\overline{\gamma}_T \leq \gamma\_{\overline{T}}.$$
>
>  Furthermore, if $|S_t|=K$ and $\mathcal{X}\_t=\mathcal{X}$, for all $t\in [T]$, then we
>
> $$\frac{1}{K}\gamma\_{KT} \leq \overline{\gamma}\_{T} \leq \gamma\_{KT}.$$
>
> Note that by relating the bound to the classical information gain, we can use lower and upper bounds derived for classical information gain for different types of kernels. In the end, we can show optimality of the regret bounds in time-dependent quantities. Such bounds are given in the new Corollary 1 in the revised manuscript, and repeated below for your convenience:
>
>
> **Corollary 1**
> Let $\delta \in (0,1)$, $T,K\in \mathbb{N}$ and let $\mathcal{X} \subset \mathbb{R}^D$ be compact and convex. Under the conditions of Theorem 3 and for the following kernels, the total regret incurred by TCGP-UCB in $T$ rounds is upper bounded (up to polylog factors) with probability at least $1-\delta$ as follows:
>
> - For the linear kernel we have: $R(T) \leq \tilde{O} \left( \sqrt{\lambda^*(K)DKT}\right) $.
> - For the RBF kernel we have: $R(T) \leq \tilde{O} \left(  \lambda^*(K)\sqrt{DKT}\right)$.
> - For the Matérn kernel we have: $R(T) \leq \tilde{O}  \left( \lambda^*(K) T^{ (D+\nu )/(D+2\nu )}\right)$,
> 	where $\nu >1$ is the Matérn parameter.

---

### Review · Reviewer_qMp3 · 2023-03-18

**Summary Of Contributions:**

This paper considers the problem of contextual combinatorial GP bandits with group constraints. Motivated by an example arising from federated learning, the authors consider a multi-armed bandit problem with changing arm sets, where in each round the agent chooses a “super-arm” to pull. The super-arm is a subset of the arms, and the agent observes rewards for each arm in the subset along with the reward of the super arm. Their goal is to a) minimize the total regret of pulling sub-optimal super arms, b) minimize the regret that arises from not satisfying “group constraints”. Groups are also subsets of the arms, and the agent needs to ensure that the reward received from various intersections between groups and subsets are above a threshold. The authors define the problem, provide an algorithm and some theory, and then an experiment.

**Audience:**

Yes

**Claims And Evidence:**

Yes

**Requested Changes:**

Generally I do not feel the need when reviewing the paper to provide feedback paragraph-by-paragraph. However, in this case, the paper was incredibly challenging to read and I think that if the authors address my concerns below, it may help greatly.

List:

- 2.2
    - This section is one of the most important in the paper since it defines the problem, but it was also very challenging to understand. To even parse the problem statement, I had to refer to Chen ‘18 many times.
    - The notation of $r_1, r_2$ are just used without being properly defined. In addition $f_1$ and $f_2$ are just used as well.
    - What does the notation $x_{t,S}$ refer to? Presumably the contexts of the vectors in $S$ - but this notation is not defined.
    - In Assumption 1, 2,3,4 - you use $\mathbf{f}$. However, $f$ is the expected reward function. It is very confusing for the reader.
- 2.4.1
    - What is “a comprising group”?
    - “which satisfy”
    - Maybe use: $ G’\subset 2^{|M_t|}| \exists G, G’\subset G, …  $
    - In general I did not understand this discussion.
- 2.4.2
    - Should it be $u(f_1(x_t…))$? What is $\hat{f}_{t,1}$?
- 2.8
    - I must confess, I did not understand this section at all. The Federated learning/crowdsourcing application seems interesting - but I don’t understand what is happening in this section. Firstly, what is a “client request pair”? I Googled this term and failed to find it. Also, what is “information leakage”? How about “data set usage”? Is it possible to give an application which is maybe a little less technical and easier to understand? There also seems to be an aspect of the problem which refers to clients leaving - is this related to the changing arm sets in the model? I only got some sense of what was happening here by looking at the setup in 5.1. You may want to reorganize this discussion.
- Section 3
    - What is the notation $[[t-1]]$?
    - What is a satisfying index?
    - In the algorithm, should line 8 be in the algorithm box?
    - Line 12 - is this where you observe $V_G$ and $r$? if so, then use this notation.
- Section 5
    - In Section 5.1: Maybe don’t use $x$ for the portion of the dataset since it is already the context?
    - Maybe in the legend of Figure 2, you can explain what the blue lines are instead of just in the caption?

**Strengths And Weaknesses:**

**Summary:** The problem seems interesting. However, my main issue with the paper is that it was very unclear in parts and difficult to read.

**Contribution:** Overall I find the contribution to be minor. The problem statement is somewhat interested (and as mentioned above related to work in Safety and bandits), but the solution is somewhat un-inspired and maybe the only obvious thing to do. In addition, the experimental section is lacking, and only consists of one baseline.  The theoretical results seem fine, but again perhaps fairly straightforward from existing results in GP bandits.

**Missing references:** This work is also to the work on safety in the multi-armed bandits literature. You may want to include citations to:

Wang, Zhenlin, Andrew J. Wagenmaker, and Kevin Jamieson. "Best arm identification with safety constraints." *International Conference on Artificial Intelligence and Statistics*. PMLR, 2022.

Amani, Sanae, Mahnoosh Alizadeh, and Christos Thrampoulidis. "Linear stochastic bandits under safety constraints." *Advances in Neural Information Processing Systems*, 32 (2019).

---

> ### Author Response · Authors · 2023-04-03
> **Response to Reviewer qMp3**
>
> **Summary:**
> *The problem seems interesting. However, my main issue with the paper is that it was very unclear in parts and difficult to read.*
>
> **Response**
> We thank the reviewer for their detailed comments. We addressed all of them and rewrote the respective portions of the paper in the revised manuscript to make it easier to read and understand.
>
> **Comment 1:**
> *Overall I find the contribution to be minor. The problem statement is somewhat interested (and as mentioned above related to work in Safety and bandits), but the solution is somewhat un-inspired and maybe the only obvious thing to do. In addition, the experimental section is lacking, and only consists of one baseline. The theoretical results seem fine, but again perhaps fairly straightforward from existing results in GP bandits.*
>
> **Response:**
> We performed new simulations on a caching-aware content recommendation setup using the MovieLens 25M dataset. We show that in this new simulation, our approach not only performs equally or better than the non-GP state-of-the-art, but that it does so while satisfying group constraints. The new simulation is given in Section 5.2 of the revised manuscript. Note that we chose to compare with only one baseline, ACC-UCB, as it is the current non-GP state-of-the-art for our setting.
>
> **Comment 2:**
> *Citation suggestions.*
>
> **Response:**
> We referenced these works related to safety in Section 1.2.
>
> **Comment 3:**
> *Multiple minor changes.*
>
> **Response:**
>
> Section 2.2:
> - The definitions of $r_1, r_2, f_1, f_2$ are given in Section 2.1 and a more detailed explanation is given in Section 2.5. We have edited both of these sections and our revisions can be seen in blue in the new manuscript.
> - We define $\boldsymbol x_{t, S}$ to be the vector of contexts of the base arms in super arm $S$ in round $t$ in Section 2.2.
> - We have changed the variable $f$ to $h$ in Assumptions 1, 2, 3 and 4.
>
> Section 2.4.1:
> - We explained the term "comprising groups" as "i.e., the groups that the base arms of the selected super arm belong to" in Section 2.4 and we modified the discussion here so that it is clearer.
>
> Section 2.4.2:
> - $\hat{f}\_{t,1}$ is the estimate of the base arm outcome function used for computing super arm reward at round $t$. We want to learn the two-output base arm outcome function $\boldsymbol f$ where the first outcome is used for super arm reward computation, and the second outcome is used for group reward computation. We denote that $\hat{f}\_{t,1}$ is the estimate of $f_1$ at round $t$.
>
> Section 2.8:
> - This section is explained more rigorously in Sections 5 and 5.1. We used section 2.8 to give the intuition.
> - In our setup we have clients, and each client has requests since we are in a federated learning application. We call these pairs as client-request pairs.
> - Context is the data set usage percentage. This means the percent of local data that the client uses to train the model. Clients don't want use all of their data set while training as this may lead to more of their information to be leaked. For higher data set usage there is a higher chance of information about the client being leaked while performing the task. This is what we mean by information leakage, it is essentially a measure of how much private information of the client is being exposed. This relationship has been investigated in (Melis et al., 2019) and we refer to it in more detail in Section 5.1.
> - We included another application of our problem that is less technical and easier to understand which is movie recommendation with content caching, more detailed information can be found in Sections 2.8 and 5.2.
> - Yes, "clients could be leaving" refers to the changing arm sets in the model.
>
> Section 3:
> - The reason why we use double bracket notation is that we have used the single bracket notation when defining "$[T] := \{1,\ldots,T\}$." So, we needed another notation to denote the posterior update at round $t-1$.
> - Satisfying name comes from satisficing which is the learner’s desire to have a reward above a threshold which was first proposed by (Reverdy et al., 2017). Please see related work. We call $i'\_{t}(x\_{t,m})$ as the satisfying index of base arm $m$ as it is used as an input for the group oracle to return the groups whose expected rewards are greater than their respective thresholds among the feasible group set. Since we apply the notion of satisficing for groups we call their indices satisfying index.
> - We fixed Lines 8 and 12 accordingly to the reviewer's comment.
>
> Section 5:
> - The reason why we use $x$ for the portion of the data set is that it is the context in this setup. Our context $x$ is the percentage of the data that is used by the client for training.
> - In Figure 2, orange refers to group regret and blue refers to super arm regret as seen in y axis. The legend is only to distinguish the line and the dashed line (i.e., color is unimportant) which are ACC-UCB and STCGP-UCB, respectively.

---

> > ### Comment · Reviewer_qMp3 · 2023-04-15
> > **Changes**
> >
> > Thanks for your edits. The federated learning motivation is a bit easier to udnerstand now.

---

### Decision · Action_Editors · 2023-06-02

**Recommendation:** Accept with minor revision

**Comment:**

The reviewers seem to be happy with the effort the authors put in addressing their concerns. The quality of the paper has improved compared to the initial submission. I agree with the reviewers that from a bandit perspective the contribution of the paper is minor and adding more baselines can strengthen the experimental section. However, the reviewers found the overall quality of the paper satisfactory and they think it could be useful for TMLR audience. I vote for the acceptance and strongly recommend the authors to ensure that they address all the reviewers' comments in the final version of the paper.

**Audience:**

The findings of this paper are relevant to a subset of TMLR's audience.

**Claims And Evidence:**

The claims made in the submission are supported by convincing evidence.